# Innovative Analogs of Unpasteurized Kombucha Beverages: Comparative Analysis of Mint/Nettle Kombuchas, Considering Their Health-Promoting Effect, Polyphenolic Compounds and Chemical Composition

**DOI:** 10.3390/ijms25147572

**Published:** 2024-07-10

**Authors:** Patrycja Pawluś, Joanna Kolniak-Ostek

**Affiliations:** Department of Fruit, Vegetable and Plant Nutraceutical Technology, Wrocław University of Environmental and Life Sciences, 37 Chelmonskiego Street, 51-630 Wroclaw, Poland; 118809@student.upwr.edu.pl

**Keywords:** *Mentha × piperita* L., *Urtica* L., phenolic compounds, bioactive compounds, SCOBY, antioxidant capacity, antidiabetic activity, anti-inflammatory activity, anticholinergic properties, UPLC/DAD/qTOF-MS/MS

## Abstract

Increasing demand for functional beverages is attracting consumers’ attention and driving research to expand our knowledge of fermentation using symbiotic culture of bacteria and yeast (SCOBY) and demonstrate the health effects of consuming kombucha. The objective of this study was to develop innovative recipes for unpasteurized mint/nettle kombucha analogs, and to compare the products obtained under varying conditions in terms of chemical composition, bioactive polyphenols and health-promoting activity. Four variants of kombucha beverages (K1–K4), differing in the addition of sucrose and fermentation temperature, were formulated. The fermentation process provided data indicating the increase of antidiabetic, anti-inflammatory and anticholinergic properties, while a decrease in antioxidant capacity was observed. The content of polyphenolics was the highest on the seventh day of fermentation. A higher fermentation temperature and a larger amount of sucrose accelerated the fermentation process, which may be crucial for shortening the production time of kombucha drinks.

## 1. Introduction

Fermentation has been a fundamental practice in global cultures for millennia. Originating from northeastern China, kombucha spread as a remedy along trade routes to Eastern Europe and Russia [1]. At present, kombucha is the fastest-growing category within the functional products. It has also become one of the most widely embraced low-alcohol, fermented beverages across the globe [2]. According to market estimates, kombucha is a fermented beverage that will be valued at 10.45 billion dollars in 2027 [3].

Kombucha, a blend of tea—primarily black but also green or oolong—and sucrose, ferments through the symbiotic culture of bacteria and yeasts (SCOBY) within a cellulose membrane. This interaction results in a buoyant mat atop the tea, with each round of successful fermentation adding a new layer [4]. The tea fungus is a colony of living organisms, wherein a symbiotic relationship between acetic acid bacteria and yeast fungi has been observed. The most commonly encountered microorganisms in kombucha are *Komagataeibacter saccharivorans* bacteria, *Saccharomyces cerevisiae*, and *Brettanomyces bruxellensis* yeasts [5]. The beverage is carbonated and possesses a mildly acidic taste [2].

Proper preparation of the tea fungus extract is crucial in terms of health safety. Kombucha fermentation is an aerobic process occurring at the liquid’s surface in the presence of oxygen. In the extract of the tea fungus, various compounds are found, including ethyl alcohol, organic acids, vitamin C, trace amounts of vitamin D, thiamine, and caffeine. Additionally, a multitude of components, such as cellulose-based polysaccharides, aldehydes, lipophilic substances, glucosides, and several enzymes [6], are present.

This golden-hued beverage is associated with many health properties, including anti-inflammatory, antioxidant, anticancer, antidiabetic, and antibacterial activities [2]. The antibiotic properties of the tea fungus extract are attributed to medusin, a natural antibiotic produced by the fungus itself [7], and various other bioactive compounds or metabolites synthesized during the fermentation [8]. When consumed in moderate amounts, it may exhibit health-promoting effects.

Further research is required to identify the composition of bioactive chemicals and conduct further in vivo tests to evaluate bioavailability. Properly controlled environmental conditions are essential for the production of kombucha, as unsterile conditions may compromise safety standards [9].

*Mentha × piperita* L. is characterized by antioxidant and anti-inflammatory properties, antimicrobial actions, and anticancer activities [10]. Extracts from peppermint are used in the treatment of intestinal colic, digestive disorders, gallbladder-related ailments, stomach ulcers, and other gastrointestinal diseases. Peppermint is widely used in the treatment of inflammatory conditions, bronchitis, toothaches, spasms, fever, headaches, and sore throats [11].

Nettle (*Urtica dioica* L.) is recognized as a rich source of phenolic compounds, which are beneficial for human health [12]. Chemical analysis of nettle has revealed various classes of compounds including terpenes, metals, vitamins, fatty acids, carotenes, polyphenolic compounds, amino acids, and others. Additionally, quantitative analysis across all samples detected compounds such as syringic acid, myricetin, quercetin, kaempferol, rutin, ellagic acid, isorhamnetin, p-coumaric acid, ferulic acid, and naringin. Glycosides and aglycones were also identified [13].

The purpose of this study was to assess the possibility of making novel, unpasteurized kombucha analogs, made from mint/nettle infusions. Additionally, the study aimed to analyze their chemical composition, polyphenolic compounds and bioactive properties, and compare the products obtained under varying conditions.

## 2. Results and Discussion

### 2.1. Basic Chemical Composition

#### 2.1.1. pH

One of the key parameters monitored during the fermentation process of various substances, including drinks, is pH. Changes in the pH value reflect the course of biochemical processes occurring under the influence of fermentation microorganisms. During the fermentation phase, the generation of organic acids contributes to a decrease in the tea’s pH level, inducing an acidic environment that diminishes oxygen availability. This condition curtails the proliferation of potential pathogenic microorganisms, thereby rendering the beverage microbiologically safe for intake, notwithstanding its biological provenance [14].

During the fermentation process, a decrease in pH value was observed (Table 1).

Variants K1 and K2 show similar pH changes, which suggests that the fermentation temperature (20 °C), regardless of the different amounts of sucrose used to prepare the fermentation mixture, is more important in terms of acidity. The K3 variant shows a more intense drop in pH compared to the K1 and K2 variants, which may indicate a more intense fermentation process related to the increased temperature (37 °C). The K4 variant shows the greatest drop in pH, which suggests the most intense fermentation process. This phenomenon is predictable due to the use of a larger amount of sucrose (15%) to prepare the mixture, and a higher fermentation temperature (37 °C) (Table 1).

Analyzing the results of pH measurements, it was found that as the kombucha drink ferments, the pH systematically decreases. After acidifying the mixture with mature kombucha, a decrease from pH ≈ 7.7 to pH ≈ 4.5 was observed, which may indicate the microbiological stability and safety of the drink. To minimize microbial contamination and acidosis, the pH of kombucha must be checked during the fermentation process, and it cannot be less than 2.5 or more than 4.2. In the K4 kombucha variant, on the 11th day of fermentation, the pH reached 2.5. This suggests that an increased sucrose content in the fermentation mixture and a higher fermentation temperature cause a more intense fermentation process due to the faster action of fermentation microorganisms. In this case, fermentation should not be prolonged.

Few studies present the risks of consuming kombucha [2]. Kombucha may be perceived as a threat to life, especially in the immunodeficient population. Kombucha consumption may be associated with acute renal failure, hyperthermia, and lactic acidosis [15]. Due to the fact that excessive consumption of kombucha may be associated with health risks, the Centers for Disease Control and Prevention stated that daily consumption of 100 g of the drink does not pose any health risk. Potentially at-risk populations include pregnant women and those sensitive to alcohol [16].

In the research of Fonteles et al. [17], the pH of kombucha decreased during the fermentation process from 4.10 on day 0 to 2.98 on day 12. For safety reasons, the time when the pH drops below 4.6 is important. In research by Hammel et al. [18] on how temperature and pH affect kombucha’s food safety, the pH fell linearly over the course of 120 h as the kombucha fermented at room temperature., reaching a value below 4.6 within 12 h. By lowering the pH, a beneficial effect is observed: bacteria causing foodborne diseases are not able to survive and multiply in kombucha.

#### 2.1.2. Organic Acids

The fermentation time depends on the desired characteristics of the drink, and its extension lowers the pH and may lead to the so-called “tea vinegar”. Increasing the degree of acidity in samples may be the main indicator of the degree of fermentation of the drink. As a result of these changes, significant amounts of acids are released in the infusion [19]. The kombucha fermentation process is carried out by a complex microbiological community, which includes acetic acid bacteria and lactic acid bacteria. Under aerobic circumstances, yeast cells catalyze the transformation of sucrose into monosaccharides, ethanol, carbon dioxide, and organic acids. Subsequently, these compounds undergo further catabolism by acetic acid bacteria, yielding acetaldehyde and acetic acid via aerobic biochemical reactions [17]. The biosynthesis of ethanol and acetic acid functions as an antimicrobial agent, suppressing the development of potentially harmful bacterial colonies within the kombucha culture [14]. Analyzing the results of the experiment, during the fermentation process, an increase in acid accumulation was observed (Table 1).

In particular, an increase in oxalic acid was noticeable after the 11th day of fermentation. In the initial phase of fermentation, the amounts of oxalic acid were lower than the amounts of malic acid in the K1, K2 and K3 variants, while in the K4 variant they had similar values. However, from the seventh day of fermentation, a predominance of oxalic acid over malic acid was observed. On the last day of the process in the K1 and K3 variants, the oxalic acid concentration was about 1.0 g/100 mL, while the K2 and K4 variants showed a concentration of approximately 1.6 g per 100 mL (Table 1). These results suggest that a larger amount of sucrose in the batch results in higher production of acids during the fermentation process, in particular oxalic acid. Throughout the entire fermentation process, acetic acid had the lowest level among the acids measured.

Compared to the values reported by Martínez Leal et al. [14] where 11 g/L of acetic acid were obtained after 30 days of the process, the concentrations of acetic acid obtained on the 16th day in the present experiment (K1–K4) were much lower, ranging between 0.09 and 0.49 g/100 mL. Much higher concentrations of acetic acid were also obtained in the study described by Shahbazi et al. [20]. Acetic acid was found to be dominant in all samples, with the highest concentration of acetic acid observed in cinnamon-flavored kombucha. There was a noticeable rise in the amount of acetic acid during fermentation, with initial and final concentrations of 737.84–982.91 mg/L on day 0 and 1131.8–2675.36 mg/L on day 16, respectively. In the present experiment, high contents of oxalic and malic acid were obtained on the 16th day of fermentation compared to literature data.

#### 2.1.3. Reducing Sugars

The maturation timeline for kombucha can span anywhere between 3 and 60 days, varying with traditional methodologies. Utilizing sucrose as the principal carbon substrate at concentrations ranging from 5% to 20%, it furnishes both the medium and the essential nutrients required for microbial growth [14]. During fermentation, significant changes occur under the influence of the activity of microorganisms, mainly bacteria and yeast. Central to these biochemical conversions is the enzymatic cleavage of sucrose into glucose and fructose, mediated by yeast-originated invertase. This enzyme facilitates the hydrolytic division of sucrose, yielding glucose and fructose, which subsequently engender ethanol through the glycolytic sequence. Through these transformations, microorganisms also produce a number of compounds that influence the taste, aroma, and nutritional properties of the final product [17].

In all kombucha variants, a dynamic decrease in sucrose was observed from the second to the fourth day of the process, and then by the seventh day the decrease was less intense (Table 1).

In the K1 and K3 variants, complete disappearance of sucrose was observed on the seventh day. This proves that the microorganisms used the entire sucrose content by the seventh day of fermentation. In the K2 and K4 variants this phenomenon occurred on the 14th day of the process. Using 15% instead of 10% sucrose to prepare the mixture extends the time in which microorganisms are able to metabolize sucrose. In each kombucha variant, the highest fructose concentration was observed on the seventh day of fermentation, followed by a gradual decrease in this parameter. The increase in glucose concentration was continuous in all kombucha variants throughout the fermentation period (Table 1).

In the study described by Gaggìa et al. [21], the sugar content in green tea-based kombucha decreased over time in all kombuchas, and glucose and fructose concentrations were found to increase during fermentation.

In the case of glucose, the results obtained in the experiment indicate a significant increase in its concentration as the fermentation time progresses (Table 1). An increase in glucose concentration during fermentation is a normal phenomenon and indicates the transformation of sucrose, which is a desirable phenomenon. The relative abundance of *Komagataeibacter* correlates with low fructose and high glucose concentrations. This may suggest that fructose is used before glucose in kombucha [22], which explains the observed results in the experiment.

### 2.2. Identification and Quantification of Phenolic Compounds

Polyphenols are key food ingredients that have complex effects on food products. On the one hand, they influence antioxidant activity, protecting the body against oxidative stress. On the other hand, they influence the taste and color of food products, which is important for their acceptance by consumers [23]. Scientific research has shown that the consumption of food rich in polyphenols can contribute to the prevention of many civilization-related diseases [24]. Nutrients with biological activity markedly reduce the likelihood of developing chronic health conditions such as atherosclerosis, diabetes mellitus, cataracts, Parkinson’s syndrome, and Alzheimer’s disease [25].

Table 2 shows a list of the 38 polyphenolics identified in a fresh infusion prepared from a mixture of peppermint and nettle (1:1). The detected chemicals were in the group of phenolic acids (17 compounds), flavonols (3 compounds), flavones and flavanones (9 compounds) and flavanols (9 compounds).

The group of phenolic acids included derivatives of gallic, caffeic, quinic, *p*-coumaric, sinapic, and protocatechuic acids (Table 2). Salvianolic acid B with [M–H]^−^ at *m/z* 717.2070 and rosmarinic acid with [M–H]^−^ at *m/z* 359.1883 were previously detected in mint samples [26,27]. Identified flavonols—(+)-gallocatechin with [M–H]^−^ at *m/z* 305.1048, (+)-gallocatechin 3-O-gallate with [M–H]^−^ at *m/z* 457.1108 and (−)-epigallocatechin with [M–H]^−^ at *m/z* 305.1084—are compounds characteristic of nettle [23]. Luteolin and apigenin and derivatives were previously detected in mint samples [21], while neoeriocitrin with [M–H]^−^ at *m/z* 595.2211 was identified in research conducted by Chou et al. [27]. Apigenin hexoside was identified in a study by Elez Garofulić et al. [28] in nettle leaves. Quercetin and isorhamnetin derivatives belonging to the flavanol group are compounds derived from both nettle [29] and mint [26]. Dihydromyricetin [M–H]^−^ at *m/z* 465.1606 was previously detected in mint by Chou et al. [27].

Table 3 and Table 4 present the results of changes in the concentration of individual phenolics in fresh mint/nettle infusion before addition of sucrose, and in the tested kombucha analogs (K1–K4) during the 16-day fermentation.

The fresh infusion contained 128.03 mg of phenolic compounds per 100 mL. The dominant group of compounds comprised flavones and flavanones, the sum of which was 104.13 mg/100 mL (Table 4). In this group, neoeriocitrin was predominant, with a content of 50.23 mg/100 mL. Phenolic acids were the second most abundant group of compounds (22.18 mg/100 mL) (Table 3). The dominant compound was caffeoylquinic acid (5.18 mg/100 mL). Flavonols constituted the third largest group, in terms of quantity, and their sum amounted to 3.90 mg/100 mL of infusion (Table 3).

Quercetin-7-*O*-[3-hydroxy-3-methylglutaroyl]hexoside was found in the highest concentration (1.87 mg/100 mL). The least numerous group of polyphenolic compounds was flavanols, the sum of which had a concentration of 1.72 mg/100 mL (Table 4). The dominant compound in this group was (+)-gallocatechin-3-*O*-gallate, the content of which was 0.75 mg/100 mL.

The fermentation process did not affect the ratio of individual groups of phenolic compounds to each other. After 16 days of fermentation, the dominant group of polyphenols comprised flavones and flavanones, the average content of which was 79.44% of all polyphenols. The second group in terms of quantity—phenolic acids—after fermentation accounted for an average of 15.98%. In all analyzed kombucha variants, an increase in the concentration of polyphenolic compounds after fermentation was observed. In the case of the K1 and K3 variants, in which 10% sugar was added, the content of polyphenol compounds increased on average by 22.5%, while in the K2 and K4 variants (15% sucrose), the polyphenol content increased on average by 84.3%. Additionally, it was observed that the highest content of polyphenolic compounds was in drinks on the seventh day of fermentation (Table 3 and Table 4). These results suggest that in order to obtain a drink with the highest polyphenol content, it is worth ending the fermentation on the seventh day.

In the research by de Oliveira et al. [2] in SCOBY-fermented green and black tea beverages over a 10-day period, approximately 127 phenolic compounds were identified, consisting of 18.3% phenolic acids, 70.2% flavonoids, 8.4% other polyphenols, 2.3% lignans, and 0.8% stilbenes. Prolonged fermentation leads to higher levels of phenolic compounds, as well as compounds such as acetic acid, malic acid, tartaric acid and B-group vitamins [2].

The analyzed samples showed a very high content (on average 65 mg/100 mL) of a compound called neoeriocitrin compared to other compounds (Table 4). Neoeriocitrin pretreatment of pancreatic beta cells with INS-1E demonstrated a protective effect against oxidative stress in a study reported by Aldemir et al. [30]. As a flavonoid, neoeriocitrin may be a possible option for an antidiabetic medication due to its antioxidant properties. Beyond pharmacological interventions, botanical antidiabetic agents are increasingly employed as complementary therapeutic options [30].

On the last day of fermentation, the amounts of polyphenols in samples K1 and K3 were 234.57 and 236.63 mg/100 mL, respectively. The equal amount of sucrose, i.e., 10%, which was used in both cases had a potential impact on the results obtained. There were differences between the samples in the fermentation temperature, which did not have a significant impact on the polyphenol content. A similar situation occurred in samples K2 and K4, where an identical amount of polyphenols was identified, i.e., 156 mg/100 mL. In both of these trials, different fermentation temperatures and the same amount of sucrose, i.e., 15%, were also used. The fermentation temperature does not directly correlate with the polyphenol content in the final product. However, there is a correlation between the sugar content used in the preparation and the amount of polyphenols.

### 2.3. Biological Activity

The defensive properties of kombucha are primarily attributed to the action of substances synthesized throughout the fermentation process, coupled with the collective influence of diverse constituents present in tea [14]. In order to determine the impact of fermentation on the biological properties of kombucha drinks, antioxidant, antidiabetic, anti-inflammatory and anti-aging properties were determined in mint/nettle infusion and kombucha brews (Table 5).

Antioxidant activity was determined during each day of fermentation, while other activities were measured at the beginning (day 0), in the middle (day 7) and at the end of the process (day 16).

#### 2.3.1. Antioxidant Capacity

Analyzing the impact of the fermentation process on the value of the ABTS (2,2′-Azino-bis(3-ethylbenzothiazoline-6-sulfonic acid)) parameter, it can be concluded that the fermentation process negatively affects the value of the parameter, lowering it (Table 5). The largest decrease was recorded in the K4 variant (decrease by 15.91 µM Tx/100 mL) and in K1 (decrease by 14.56 µM Tx/100 mL), then in K3 (decrease by 11.29 µM Tx/100 mL), and in K2 (7.52 µM Tx/100 mL). This may suggest that the fermentation process conditions used in K2 (20 °C and 15% sucrose) create the most stable environment for ABTS among those used in the experiment.

The fermentation process had a negative impact on the value of the FRAP (ferric reducing antioxidant power assay) parameter in variants K1 and K2, reducing it (Table 5), in K1 by 7.72 µM Tx/100 mL and in K2 by 7.51 µM Tx/100 mL. However, in the K3 and K4 variants, the fermentation process had a positive effect on the value of the FRAP parameter, increasing its value in K3 by 5.95 µM Tx/100 mL and in K4 by 22.28 µM Tx/100 mL. This suggests that the higher fermentation temperature (37 °C) used in K3 and K4 had a positive effect on the FRAP parameter value. A greater increase in the FRAP value was observed in K4, in the variant in which a larger amount of sucrose was used to prepare the mixture (15%) than in K3 (10%). Higher temperature and sucrose content to prepare the mixture created the best conditions for increasing the FRAP parameter.

The results of the antioxidant activity determined by the DPPH (2,2-Diphenyl-1-(2,4,6-trinitrophenyl)hydrazyl) test showed that the fermentation process reduced the value of the DPPH parameter in all kombucha variants throughout the fermentation process lasting 16 days (Table 5). The DPPH value decreased to a similar extent in the kombucha variants K1 (by 18.29 µM Tx/100 mL), K3 (by 15.34 µM Tx/100 mL) and K4 (by 16.91 µM Tx/100 mL), while the smallest decrease was observed in the K2 variant, only by 3.38 µM Tx/100 mL. In variants K3 and K4, the highest values of the DPPH parameter were recorded before the fermentation process, while in variants K1 and K2 they were recorded on the fourth day of the process. This may suggest that the fermentation temperature (20 °C) is the optimal temperature for increasing the DPPH content in kombucha during the first 4 days of fermentation.

As described by Martínez Leal et al. [10], green tea-based kombucha showed the strongest activity against hydroxyl radicals, DPPH, and superoxide anions. In contrast, black tea has the highest level of activity against DPPH radicals until the fifteenth day of fermentation. The DPPH activity was in the range of 26.33–170.13 μL/mL and it was observed that it increased with the duration of fermentation, as also described in studies conducted by Vulić et al. [31]. In the present experimental kombucha drinks, the lowest DPPH value (47.37 µM Tx/100 mL) was obtained in the K1 variant on the 14th day of fermentation, while the highest (106.71 µM Tx/100 mL) was obtained in the K2 variant on the fourth day of the process (Table 5).

Olech et al. [32] found that kombucha brews had high reducing power (1092.04 to 9258.70 μmol Fe^2+^/g) and antioxidant potential (0.26–4.25 mmol Trolox/g). During kombucha fermentation, bacteria digest sugar, reducing its amount while increasing the antioxidant activity of aglycones. Aglycones’ increased antioxidant activity and lower sugar content after fermentation may improve health by controlling blood glucose levels and lowering the risk of diabetes.

#### 2.3.2. Antidiabetic Activity

Scientific research has shown that polyphenols can replace synthetic α-amylase and α-glucosidase inhibitors, and phenolic substances are adept at orchestrating the metabolic processes of lipids and carbohydrates by impeding the enzymatic action of α-amylase and α-glucosidase. The inhibition of these enzymes is facilitated by their chelating capacity, structural modification potential, and the subsequent curtailment of their biological activities. α-Amylase and α-glucosidase are integral to the metabolic processing of carbohydrates. Therefore, by inhibiting the activity of these enzymes, this will result in a diminution of the glucose incorporation rate into the systemic circulation, ultimately precipitating a decline in the glycemic levels within the bloodstream [33].

The fermentation process increased the anti-diabetic properties in all kombucha variants (Table 5). An increase in the inhibitory activity of α-amylase in K1 by 4.08%, in K2 by 6.35%, in K3 by 3.28%, and in K4 by 3.64% was observed. In the case of α-glucosidase an increase in the inhibitory activity in K1 by 5.01%, K2 by 12.07%, K3 by 2.92%, and in K4 by 4.21% inhibition was observed. In the experiment, after the fermentation process the highest antidiabetic activity was observed in variants K1 (70.21% for α-glucosidase and 83.25% for α-amylase) and K3 (68.12% for α-glucosidase and 82.45% for α-amylase). Variants K2 and K4 were characterized by slightly lower activity: for anti-α-glucosidase activity it was 65.98% and 58.12%, respectively, while for anti-α-amylase activity it was 77.49% and 74.78%, respectively. The results suggest that, taking into account antidiabetic activity, the variants with a lower amount of sucrose were characterized by higher health-promoting properties, and the variants prepared with a higher sucrose content (15%) had lower activity.

In a study by Aloulou et al. [34], rats with diabetes (induced with alloxan) were given 5 mL/kg of kombucha or black tea for 30 days to observe whether it inhibited the enzyme α-amylase. The findings indicated that the rodents consuming kombucha exhibited enhanced inhibitory responses towards the enzyme α-amylase within the pancreatic tissue and plasma, alongside improved regulation of postprandial glucose levels, in contrast to their counterparts ingesting black tea. In the study described by Geraris Kartelias et al. [33], the IC_50_ values of α-glucosidase and α-amylase inhibition in kombucha samples were 17.81–18.8 µL, and 221.87–243.61 µL, respectively. Investigations also determined that the fortification of kombucha with lavender flowers, rose petals, and hibiscus calyces considerably enhanced the suppression of the enzymes α-amylase and α-glucosidase. On the other hand, the addition of lemon peel, turmeric, and ginger reduced the suppressive effect on α-glucosidase while strengthening the inhibitory effect on α-amylase.

#### 2.3.3. Anticholinergic Activity

Consequent to the biochemical fermentation sequence, an increase in the anticholinergic effect was observed in all kombucha variants, with the highest inhibitory activities observed for the K3 variant (Table 5). Analyzing the anti-aging properties, it can be concluded that the greatest potential was detected for the variant fermented at an increased temperature (37 °C) and for which a lower amount of sucrose was used (10%).

In the study by Olech et al. [32], it was found that flower tea is a stronger inhibitor of acetylcholinesterase (61.63% enzyme inhibition) than nut tea. Rosehip extracts showed similar, moderate effects. These changes suggest the potential occurrence of positive effects of fermentation on the biological activity of various components contained in the samples. In a study by Geraris Kartelias et al. [33], it was found that by enriching kombucha with ginger, turmeric, lemon peel, lavender flowers rose petals, and hibiscus calyces the inhibitory effect against acetylcholinesterase and butyrylcholinesterase was markedly enhanced. The suppression of cholinesterase activity is a prevalent therapeutic approach for Alzheimer’s pathology. Extracts derived from Hibiscus species exhibit potent suppressive properties that could potentially enhance the management and mitigation of Alzheimer’s pathology through the attenuation of AChE and BuChE activities. Flavonoid constituents, such as quercetin and rutin, act as competitive inhibitors of AChE and BuChE.

#### 2.3.4. Anti-Inflammatory Activity

The fermentation process increased the anti-inflammatory activities of COX-1 and COX-2 in all kombucha variants (Table 5). In the present experiment, the fermentation process caused the greatest increase in the effect of COX-1 in K3 followed by, in descending order, K4, K2, K1; and of COX-2 in K2 kombucha followed by, in descending order, K3, K4, K1. The fermentation process had the least impact on anti-inflammatory activities in kombucha K1 fermented at 20 °C and for which a lower sucrose content was used (10%).

Inflammatory processes are often accompanied by persistent dilation of capillaries (increased permeability). Commonly used nonsteroidal anti-inflammatory drugs have been observed to counteract such changes in permeability, thereby inhibiting exudation and subsequent edema formation. In a study by Saha [35], tea extract was found to have significant anti-exudative properties, similar to aceclofenac, which blocks the enzyme cyclooxygenase. It is less receptive to COX-1 than COX-2. Aceclofenac suppresses the generation of several inflammatory mediators, including prostaglandins, interleukins, and tumor necrosis factors from the arachidonic acid pathway, as a result of inhibiting COX-2. As described by Sales et al. [36], the consumption of Camellia sinensis tea has a number of beneficial biological effects, such as the prevention of cardiovascular disease, colorectal cancer and type 2 diabetes. These effects mainly concern the anti-inflammatory and antioxidant activities of C. sinensis. Black tea and kombucha infusions showed a decrease in oxidative stress in HK-2 cells exposed to indoxyl sulfate and high glucose, while also lowering the amount of uric acid in the extracellular fluid. The theory proposed is that phenolic components, such as catechins, rutin, quercetin, and chlorogenic acids, possibly in combination with caffeine, are responsible for the positive effects reported with these drinks. The study also revealed that the expression of the inflammatory enzymes COX-2 and iNOS (inducible nitric oxide synthase) was decreased by an aqueous coffee cascara extract.

In a study by Xia et al. [37], it was found that combining white tea with mint increased the antibacterial and anti-inflammatory effects. This suggests that kombucha based on a mixture of white tea and mint would be a good subject for further research. An enhanced collective inhibitory impact was noted on quartet bacterial cultures, notably on Staphylococcus argenteus. Moreover, the integrative concoction exhibited more pronounced anti-inflammatory responses in biological systems than the singular application of either constituent, correlated with a diminution in the concentrations of pro-inflammatory cytokines such as inducible nitric oxide synthase (iNOS), tumor necrosis factor alpha (TNF-α), interleukin-1beta (IL-1β), interleukin-6 (IL-6), and cyclooxygenase-2.

### 2.4. Chemometric Analysis

Principal component analysis (PCA) examines the relationships between concentrations of different groups of phenolic compounds (phenolic acids, flavonols, flavones and flavanones, and flavanols), concentrations of individual sugars (sucrose, fructose, glucose), organic acids (acetic, malic and oxalic acids), and biological activities (ABTS, DPPH, FRAP—antioxidant capacity; COX1, COX2—anti-inflammatory activity; a-amylase, a-glucosidase—antidiabetic activity; AChE, BuChE—anticholinergic activity) in nettle/mint infusion (K0) and 4 types of nettle/mint kombuchas before fermentation (K1_0–K4_0) on the seventh day of fermentation (K1_7–K4_7) and on the 16th day of fermentation (K1_16–K4_16) (Figure 1a).

Two major components with eigenvalues larger than one were found during the PCA analysis, and together they accounted for 71.95% of the variation in the data. PC1 values explained 61.80% of the total variance and confirmed that fermentation increases the biological properties of kombucha drinks. The second main component (PC2) reflects the antioxidant properties of polyphenolic compounds and shows that the free radicals’ reducing ability was negatively correlated with the duration of fermentation. Samples K1 and K3 after 7 and 16 days of fermentation were characterized by a high concentration of phenolic compounds as well as strong antidiabetic properties. This was most likely due to the lower sugar addition to the samples (10%) compared to the K2 and K4 samples (15%). Strong antidiabetic properties of phenolic compounds were confirmed by the research of Huang et al. [38]. Their study identified a positive effect of chlorogenic acid in alleviating inflammatory disorders, cardiovascular anomalies, cerebrovascular diseases and diabetes. Chlorogenic acid can slow down the secretion of glucose into the circulatory system after a meal and also increases the sensitivity of cells to insulin, which is important in the context of diabetes. Chlorogenic acid also has anti-inflammatory effects.

Additionally, the biplot shows that kombuchas after 16-day fermentation are characterized by low sucrose content, which confirms its consumption by microorganisms during fermentation. Variants K2 and K4 to which 15% sucrose was added both on days 0 and 7 of fermentation were strongly related to the high sucrose content. All samples after 16 days of fermentation were characterized by a high concentration of organic acids and high anti-inflammatory properties (COX1 and COX2), as well as anticholinergic properties. This allows us to conclude that organic acids have a strong effect on inhibiting inflammation and acting against AChE and BuChE. The high biological activity of organic acids was confirmed in the research of Alakolanga et al. [39], who concluded that malic acid has strong anti-obesity and anti-diabetic properties. Additionally, the graph shows that the fermentation process at 37 °C (K2 and K4) contributed to the increase in biological properties compared to samples at 20 °C (K1 and K3).

The above-mentioned observations are confirmed by the statistical calculations. Strong positive correlations between individual organic acids and anticholinergic and anti-inflammatory activity were observed (Figure 1b). The correlation between malic acid and activity against acetylcholinesterase was r = 0.85, while against butyrylcholinesterase it was r = 0.73. For oxalic acid, these values were r = 0.87 and r = 0.72, respectively. Slightly lower correlation coefficients were obtained for acetic acid: in the case of activity against acetylcholinesterase it was r = 0.68, while against butyrylcholinesterase it was r = 0.59. There were also positive correlations between the content of individual groups of phenolic compounds and anti-diabetic activities (α-glucosidase and α-amylase), anticholinergic (acetylcholinesterase and butyrylcholinesterase) and anti-inflammatory activity (COX1 and COX2). Additionally, the graph also confirms the negative correlation between the content of polyphenolic compounds and antioxidant capacity. From these observations we can conclude that in the case of kombucha drinks made from mint/nettle infusions, their biological activity is related to chemical compounds produced by microorganisms during fermentation, which is confirmed by the research of Martínez Leal et al. [14].

## 3. Materials and Methods

### 3.1. Plant Material and Sample Processing

The research material consisted of four variants of kombucha-like beverages prepared on the laboratory scale. In the initial stage, herbal mixtures were used to prepare brews (K1–K4). A mixture of dried herbs was created by combining peppermint (*Mentha piperita* L.) in a 1:1 (*w/w*) ratio with common nettle (*Urtica dioica* L.). Dried peppermint and nettle were purchased in retail (Dary Natury Sp. z o.o., Koryciny, Poland). The raw material came from organic farming (certificate PL-EKO-01-001493) located in the Podlaskie Voivodeship in Poland. Ten grams of the herbal mixture were used per 2 L of each brew, along with an additional 250 mL of starter, which was mature kombucha.

In brews K1 and K3 10% sucrose was added, while in K2 and K4 15% sucrose was added. The brews K1 and K2 were stored at 20 °C, while K3 and K4 were stored in 37 °C incubators (POL-EKO, Wodzisław Śląski, Poland). The fermentation of all brews lasted for 16 days. Samples were analyzed after making the infusion, adding the starter, and at 2, 4, 7, 11, 14, and 16 days of fermentation.

The SCOBY culture consists of three strains of yeast (*Saccharomyces cerevisiae*, *Saccharomyces ludwigii*, *Schizosaccharomyces pombe*), two strains of acetic acid bacteria (*Acetobacter pasteurans* and *Acetobacter aceti*) and two strains of lactic acid bacteria (*Lactobacillus acidophilus* and *Lactobacillus fermentum*). The specific growth media on which the microorganisms were grown were: acetobacter agar for Acetobacter, MRS Agar for lactic acid bacteria and Sabuardo Agar with chloramphenicol for yeast.

The inoculum used in the research was kombucha obtained after 10 days of fermentation on *Camellia sinensis* infusion and distilled water with the addition of sucrose in the amount of 60 g/1000 mL. The fermentation temperature was 27 °C. Kombucha starter was added in an amount of 10% (*v/v*).

In all samples, total soluble solids, pH, content of reducing sugars and organic acids, identification and quantification of phenolic compounds, antioxidant capacity, as well as antidiabetic, anti-inflammatory and anticholinergic properties, were determined.

### 3.2. Physicochemical Analyses

The TSS (total soluble solids) content was measured with an Atago PR-101 digital refractometer (Atago Co. Ltd., Tokyo, Japan) and expressed as °Brix. pH was measured using a DL-21 automatic titrator (Mettler-Toledo, Schwerzenbach, Switzerland).

### 3.3. Determination of Sugars and Organic Acids

For the determination of the soluble sugars and organic acid content the method described by Kolniak-Ostek [40], was used. For the determination of the soluble sugar a Merck-Hitachi L-7455 liquid chromatograph was used. For organic acid composition, a Dionex (Sunnyvale, CA, USA) liquid chromatograph was used. Each set of data was acquired in triplicate, and presented in grams per 100 mL of drinks.

### 3.4. UPLC-PDA-Q/Tof-MS Analysis of Polyphenolics

Analysis of phenolic compounds from kombucha samples was carried out with a G2 Q-Tof mass detector with an ACQUITY UPLC-PDA system according to the method described by Kolniak-Ostek [40]. Polyphenols were identified using an ACQUITY UPLC system equipped with a PDA detector and a G2 Q-Tof mass detector (Waters, Manchester, UK) with ESI source operating in negative mode. Individual polyphenols were separated using a UPLC BEH C18 column (1.7 μm, 2.1 × 100 mm, Waters) at 30 °C during 15 min. The mobile phase consisted of 0.1% formic acid (*v/v*) (solvent A) and 100% acetonitrile (solvent B), with a linear gradient sequence and constant flow rates of 0.42 mL/min. The runs were monitored at the following wavelengths: 254 nm for flavonols, 280 nm for flavan-3-ols, 320 nm for phenolic acids, 340 nm for flavones, and 360 nm for flavonol glycosides. The PDA spectra were obtained in 2 nm steps over the wavelength range of 200–600 nm. The retention periods and spectra were compared to authentic standards. External calibration curves were used to quantify phenolic compounds, with reference compounds chosen based on the principle of structure-related target analyte/standard (chemical structure or functional group).

Full scan, data-dependent MS scanning from *m/z* 100–2500 was used for the analysis. At a concentration of 500 pg/mL, leucine enkephalin (*m/z* 554.2615 Da) was used as the reference substance. The optimal MS settings were as follows: 2500 V capillary voltage, 30 V cone voltage, 100 °C source temperature, 300 °C desolvation temperature, and 300 L/h desolvation gas (nitrogen) flow rate.

Experiments on collision-induced fragmentation were carried out with argon as the collision gas and voltage ramping cycles ranging from 0.3 to 2 V. The retention duration and precise molecular masses were used to characterize the individual components. Before and after fragmentation, each chemical was optimized to its predicted molecular mass in the negative mode. The UPLC-MS data were entered into the MassLynx 4.0 software (Waters, Milford, MA, USA).

Calibration curves were determined experimentally for caffeic, gallic, caffeoylquinic, protocatechuic and sinapic acids, *p*-coumaroylquinic acid, salvianolic acid B, rosmarinic acid, (+)-gallocatechin, neoeriocitrin, quercetin 3-*O*-glucoside, isorhamnetin 3-*O*-glucoside, apigenin 7-*O*-glucoside, myricetin and luteolin 3-*O*-glucoside. Caffeic acid derivatives were expressed as caffeic acid, *p*-coumaroylquinic acid derivatives were expressed as *p*-coumaroylquinic acid, gallic acid derivatives were expressed as gallic acid, gallocatechins were expressed as (+)-gallocatechin, quercetin derivatives were expressed as quercetin 3-*O*-glucoside, isorhamnetin derivatives were expressed as isorhamnetin 3-*O*-glucoside, derivatives of apigenin were expressed as apigenin 7-*O*-glucoside, and luteolin derivatives were expressed as luteolin 3-*O*-glucoside. All standards were purchased from Merck KGaA (Darmstadt, Germany).

All determinations were done in triplicate (n = 3).

### 3.5. Antioxidant Capacity

The antioxidant activity was evaluated using the ABTS, FRAP, and DPPH methods previously described by Kolniak-Ostek et al. [41] using a Synergy H1 microplate reader (BioTek, Winooski, VT, USA). All data were taken in triplicate and reported as mmol of Trolox equivalents per 100 mL of kombucha.

### 3.6. α-Amylase and α-Glucosidase Inhibition Assay

Alpha-amylase and alpha-glucosidase inhibition assays were performed according to the method described by Podsędek et al. [42]. All measurements were made in triplicate with a Synergy H1 microplate reader and expressed as % inhibition.

### 3.7. Anticholinergic Activity

Anticholinergic activity was determined as the ability to inhibit the enzymes acetylcholinesterase (AChE) and butyrylcholinesterase (BuChE) using Ellman’s method described by Gironés-Vilaplana et al. [43]. Measurements were performed in triplicate using a Synergy H1 microplate reader and expressed as % inhibition.

### 3.8. Anti-Inflammatory Activity

The anti-inflammatory action was assessed by spectrophotometric measures of the inhibition of cyclooxygenase (COX-1, COX-2), as described by Mizgier et al. [44]. Measurements were made in triplicate with a Synergy H1 microplate reader and expressed as % inhibition.

### 3.9. Statistical Analysis

The mean values (n = 3) were compared with Duncan’s test and one-way ANOVA (*p* ≤ 0.05). All statistical analyses and principal component analysis (PCA) were performed using Statistica 13.3 software (StatSoft, Kraków, Poland). The Factoextra R 1.0.7 package was used to determine the chemometric analysis.

## 4. Conclusions

The quality of kombucha drinks made from mint/nettle infusions can be optimized by the parameters of the fermentation process. Effective use of the fermentation process to obtain maximum polyphenol values in kombucha includes limiting the fermentation time to 7 days; further prolongation of fermentation causes a decrease in the amount of polyphenols. Increasing the amount of sucrose used in the mixture from 10% to 15% leads greater production of acids during fermentation, especially oxalic acid. This indicates that there is a relationship between the amount of sugar and the amount of acids produced. An increased fermentation temperature and a larger amount of sucrose speed up the fermentation process, which may be crucial for shortening the production time of a kombucha drink.

The bioactive qualities of kombucha beverages and organic acids showed a significant positive association, as demonstrated by chemometric investigations. The results of the analyses verified the hypothesis that kombucha-type drinks with high bioactive properties can be obtained from raw materials other than tea, and the appropriate selection of sucrose content and process temperature can determine the biological features of kombucha. Based on the present results we can conclude that, for products with higher biological activity, more research is needed on the raw materials utilized, the conditions of the fermentation process, and the interactions between the ingredients of kombucha drinks and their effect on the drinks’ quality.

## Figures and Tables

**Figure 1 ijms-25-07572-f001:**
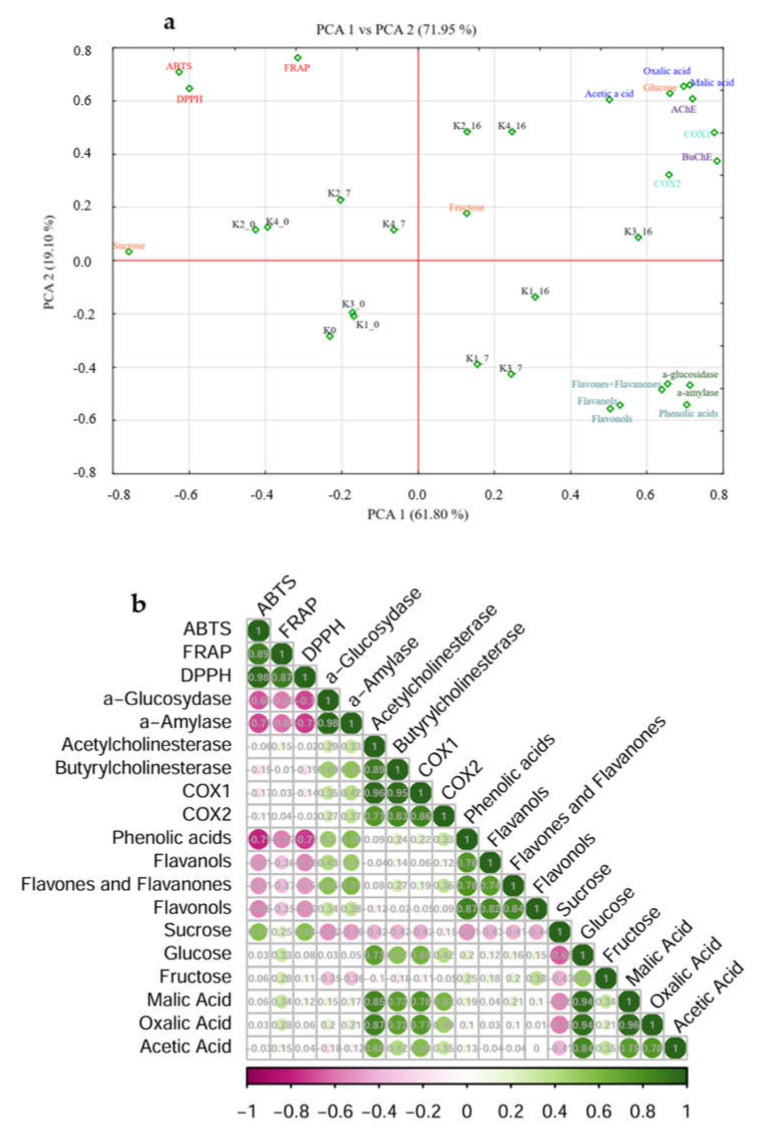
(**a**) Principal component analysis, a score plot of the mint/nettle kombuchas’ first two principal components. The relationship between the kombucha’s chemical composition and biological activity is depicted by the PCA score plot; (**b**) Each principal component is given a physical meaning through the plot of normalized factor loadings. The loadings show the degree of correlation between the two variables (individual phenolic groups, organic acids, reducing sugars, and biological activities) and certain principal components (PCs). Abbreviations: K0—fresh mint/nettle infusion; K1—kombucha with 10% of sugar addition, fermented in 20 °C; K2—kombucha with 15% of sugar addition, fermented in 20 °C; K3—kombucha with 10% of sugar addition, fermented in 37 °C; K4—kombucha with 15% of sugar addition, fermented in 37 °C; _0—0 day of fermentation; _7—7th day of fermentation; _16—16th day of fermentation.

**Table 1 ijms-25-07572-t001:** pH value, organic acids (mg/100 mL) and reducing sugars (mg/100 mL) in mint/nettle fresh infusion and kombucha brews during fermentation (average ± standard deviation; n = 3).

			Organic Acids			Reducing Sugars		
		pH	Malic Acid	Oxalic Acid	Acetic Acid	Sucrose	Glucose	Fructose
Fresh infusion		7.78 ± 0.04 ^a^	0.00 ± 0.00 ^l^	0.00 ± 0.00 ^n^	0.00 ± 0.00 ^n^	0.00 ± 0.00 ^d^	0.00 ± 0.00 ^o^	0.00 ± 0.00 ^l^
K1	Day 0	4.12 ± 0.03 ^c^	0.15 ± 0.02 ^k^	0.00 ± 0.00 ^n^	0.00 ± 0.00 ^n^	9.40 ± 0.12 ^e^	0.14 ± 0.01 ^n^	0.13 ± 0.01 ^k^
	Day 2	3.43 ± 0.04 ^de^	0.21 ± 0.02 ^j^	0.11 ± 0.01 ^mn^	0.01 ± 0.00 ^mn^	7.70 ± 0.13 ^f^	1.12 ± 0.01 ^m^	0.78 ± 0.02 ^i^
	Day 4	3.56 ± 0.03 ^d^	0.29 ± 0.01 ^h^	0.18 ± 0.01 ^lm^	0.04 ± 0.00 ^lm^	6.94 ± 0.08 ^g^	2.15 ± 0.02 ^k^	1.95 ± 0.03 ^f^
	Day 7	3.24 ± 0.00 ^fg^	0.32 ± 0.00 ^gh^	0.39 ± 0.01 ^j^	0.04 ± 0.00 ^lm^	1.80 ± 0.15 ^kl^	3.23 ± 0.01 ^i^	2.32 ± 0.02 ^de^
	Day 11	3.08 ± 0.02 ^g–j^	0.37 ± 0.01 ^g^	0.48 ± 0.03 ^ij^	0.05 ± 0.01 ^kl^	0.00 ± 0.00 ^o^	3.35 ± 0.02 ^i^	1.86 ± 0.01 ^fg^
	Day 14	2.96 ± 0.01 ^i–k^	0.44 ± 0.02 ^f^	0.67 ± 0.02 ^g^	0.06 ± 0.01 ^jk^	0.00 ± 0.00 ^o^	4.04 ± 0.02 ^g^	1.12 ± 0.01 ^h^
	Day 16	2.94 ± 0.01 ^jk^	0.57 ± 0.01 ^d^	0.98 ± 0.03 ^e^	0.09 ± 0.00 ^hi^	0.00 ± 0.00 ^o^	4.86 ± 0.03 ^f^	0.66 ± 0.01 ^ij^
K2	Day 0	4.21 ± 0.02 ^bc^	0.00 ± 0.00 ^l^	0.00 ± 0.00 ^n^	0.00 ± 0.00 ^n^	15.05 ± 0.21 ^a^	0.25 ± 0.01 ^n^	0.18 ± 0.01 ^k^
	Day 2	3.48 ± 0.02 ^d^	0.31 ± 0.01 ^gh^	0.15 ± 0.02 ^m^	0.02 ± 0.00 ^mn^	12.55 ± 0.15 ^b^	1.59 ± 0.01 ^lm^	1.21 ± 0.01 ^gh^
	Day 4	3.50 ± 0.01 ^d^	0.45 ± 0.01 ^e^	0.25 ± 0.01 ^l^	0.05 ± 0.01 ^kl^	11.38 ± 0.18 ^c^	3.45 ± 0.02 ^hi^	3.09 ± 0.02 ^c^
	Day 7	3.25 ± 0.01 ^fg^	0.48 ± 0.02 ^e^	0.50 ± 0.02 ^i^	0.06 ± 0.01 ^jk^	5.34 ± 0.22 ^h^	4.97 ± 0.02 ^f^	3.79 ± 0.02 ^c^
	Day 11	3.14 ± 0.00 ^f–i^	0.59 ± 0.02 ^cd^	0.75 ± 0.01 ^f^	0.08 ± 0.01 ^ij^	2.18 ± 0.19 ^jk^	5.54 ± 0.03 ^e^	3.12 ± 0.01 ^c^
	Day 14	3.05 ± 0.02 ^h–j^	0.70 ± 0.02 ^b^	1.14 ± 0.03 ^d^	0.11 ± 0.01 ^fg^	1.03 ± 0.01 ^m^	6.16 ± 0.02 ^de^	2.43 ± 0.01 ^d^
	Day 16	2.99 ± 0.01 ^h–j^	0.91 ± 0.01 ^a^	1.54 ± 0.02 ^b^	0.15 ± 0.01 ^e^	0.00 ± 0.00 ^o^	7.14 ± 0.03 ^c^	1.52 ± 0.01 ^g^
K3	Day 0	4.36 ± 0.03 ^b^	0.20 ± 0.00 ^l^	0.10 ± 0.00 ^n^	0.02 ± 0.00 ^n^	9.40 ± 0.12 ^e^	0.27 ± 0.01 ^n^	0.15 ± 0.01 ^k^
	Day 2	3.28 ± 0.02 ^ef^	0.27 ± 0.01 ^i^	0.16 ± 0.01 ^m^	0.05 ± 0.00 ^kl^	7.50 ± 0.20 ^f^	1.56 ± 0.01 l^m^	1.02 ± 0.01 ^hi^
	Day 4	3.16 ± 0.02 ^f–h^	0.36 ± 0.01 ^g^	0.21 ± 0.01 ^l^	0.10 ± 0.01 ^gh^	6.72 ± 0.11 ^g^	2.65 ± 0.02 ^jk^	2.13 ± 0.02 ^ef^
	Day 7	2.81 ± 0.03 ^kl^	0.45 ± 0.02 ^ef^	0.44 ± 0.02 ^j^	0.12 ± 0.00 ^ef^	1.00 ± 0.03 ^m^	3.61 ± 0.02 ^h^	2.40 ± 0.01 ^d^
	Day 11	2.72 ± 0.01 ^lm^	0.49 ± 0.02 ^e^	0.50 ± 0.02 ^i^	0.15 ± 0.01 ^e^	0.00 ± 0.00 ^o^	4.01 ± 0.01 ^g^	1.76 ± 0.01 ^f^
	Day 14	2.70 ± 0.01 ^l–n^	0.53 ± 0.03 ^d^	0.60 ± 0.02 ^h^	0.18 ± 0.01 ^de^	0.00 ± 0.00 ^o^	4.90 ± 0.02 ^f^	1.02 ± 0.02 ^hi^
	Day 16	2.72 ± 0.00 ^lm^	0.74 ± 0.03 ^b^	1.01 ± 0.03 ^de^	0.20 ± 0.01 ^d^	0.00 ± 0.00 ^o^	5.23 ± 0.02 ^ef^	0.54 ± 0.02 ^i^
K4	Day 0	4.34 ± 0.01 ^b^	0.23 ± 0.02 ^j^	0.20 ± 0.01 ^l^	0.10 ± 0.01 ^gh^	15.05 ± 0.21 ^b^	0.99 ± 0.01 ^m^	0.31 ± 0.02 ^j^
	Day 2	3.17 ± 0.02 ^f–h^	0.31 ± 0.02 ^gh^	0.35 ± 0.01 ^k^	0.13 ± 0.00 ^ef^	12.31 ± 0.12 ^b^	2.01 ± 0.01 ^k^	1.55 ± 0.02 ^g^
	Day 4	3.04 ± 0.02 ^h–j^	0.41 ± 0.01 ^f^	0.41 ± 0.02 ^j^	0.19 ± 0.01 ^de^	10.89 ± 0.25 ^d^	3.89 ± 0.02 ^gh^	4.14 ± 0.01 ^b^
	Day 7	2.61 ± 0.01 ^m–o^	0.50 ± 0.02 ^de^	0.52 ± 0.01 ^i^	0.25 ± 0.01 ^d^	4.14 ± 0.02 ^i^	5.39 ± 0.02 ^e^	5.02 ± 0.03 ^a^
	Day 11	2.50 ± 0.01 ^o^	0.56 ± 0.01 ^d^	0.69 ± 0.02 ^fg^	0.35 ± 0.00 ^c^	1.31 ± 0.01 ^l^	6.69 ± 0.02 ^d^	4.01 ± 0.02 ^bc^
	Day 14	2.46 ± 0.02 ^o^	0.64 ± 0.02 ^c^	1.36 ± 0.04 ^c^	0.41 ± 0.01 ^b^	0.52 ± 0.01 ^n^	8.14 ± 0.03 ^b^	2.46 ± 0.01 ^d^
	Day 16	2.53 ± 0.01 ^no^	0.85 ± 0.00 ^ab^	1.67 ± 0.04 ^a^	0.49 ± 0.01 ^a^	0.00 ± 0.00 ^o^	9.55 ± 0.02 ^a^	1.65 ± 0.02 ^fg^

Means of three independent analyses ± standard deviation; Values in the same columns followed by different letters are significantly different at *p* < 0.05 according to Duncan’s test. Abbreviations: K1—kombucha with 10% of sugar addition, fermented in 20 °C; K2—kombucha with 15% of sugar addition, fermented in 20 °C; K3—kombucha with 10% of sugar addition, fermented in 37 °C; K4—kombucha with 15% of sugar addition, fermented in 37 °C.

**Table 2 ijms-25-07572-t002:** Retention times, UV–vis spectra and characteristic ions of phenolic compounds of mint/nettle fresh infusion.

Rt (min)	λmax (nm)	[M–H]^−^ (*m/z*) ^1^	MS/MS Fragments (*m/z*) ^1^	Tentative Identification
Phenolic acids and derivatives				
1.56	321	169.0464	125.0518	Gallic acid ^2^
1.76	320	179.0825	161.0437/135.0587	Caffeic acid ^2^
2.48	324	315.1491	153.0766	Protocatechuic acid hexoside
2.59	320	153.0444	109.0470	2,3-Dihydrobenzoic acid
2.80	324	483.0938	331.0939/169.0304	Digalloylglucose
3.12	320	353.1337	191.0854/179.0633/135.0734	Caffeoylquinic acid ^2^ (isomer I)
3.19	320	355.0831	223.0434	Sinapic acid pentoside
3.55	321	321.0387	169.0304	Digallic acid
3.78	320	495.1649	309.1016	3,5-Digalloylquinic acid
3.83	322	337.1338	175.0886/163.0684	*p*-Coumaroylquinic acid ^2^
3.95	324	483.0938	321.0387/169.0304	Digalloylglucose
4.03	326	353.1337	191.0882	Caffeoylquinic acid (isomer II)
4.30	326	353.1299	173.0762	Caffeoylquinic acid (isomer III)
4.42	324	341.1529	179.0661/135.0710	Caffeoyl hexoside
5.33	314	295.0826	115.0252	*p*-Coumaroyl tartaric acid
7.84	325	717.2070	519.1472/321.0827/295.1001	Salvianolic acid B ^2^
8.12	328	359.1183	197.0776/179.0606/161.0538	Rosmarinic acid ^2^
Flavanols				
3.70	279	305.1048	261.2162/219.1045	(+)-Gallocatechin ^2^
4.95	278	305.1084	261.1141/219.0560	(+)-Epigallocatechin
5.12	280	457.1108	305.1084/169.0437	(+)-Gallocatechin 3-*O*-gallate
Flavones and Flavanones				
4.84	346	609.1523	447.1031/285.0582	Luteolin diglucoside
5.55	346	637.1675	285.0824	Luteolin diglucuronide
5.66	340	593.2099	431.1359/269.0840	Apigenin 6,8-di-*C*-glucoside
5.73	340	563.1732	431.2094/401.3017/269.0336	Apigenin *C*-hexoside-*C*-pentoside
6.45	345	433.1978	271.0702	Apigenin hexoside
6.56	355	595.2211	287.0943/151.0325	Neoeriocitrin ^2^
6.65	346	593.2049	285.0790	Luteolin-7-*O*-rutinoside
6.86	345	461.1247	285.0755	Luteolin-7-*O*-glucuronide
7.40	340	579.2217	271.1028	Apigenin-7-*O*-rutinoside (Isorhoifolin)
Flavonols				
4.14	340	465.1606	301.1234	Dihydromyricetin
4.32	350	625.1372	301.0417	Quercetin-diglucoside
5.01	351	639.1689	477.1142/315.0655	Isorhamnetin dihexoside (isomer I)
6.08	351	639.1638	314.0597	Isorhamnetin dihexoside (isomer II)
6.30	350	615.0989	301.0204	Quercetin-*O*-galloyl-glucoside ^2^
6.70	352	463.1404	300.0657	Quercetin 3-*O*-galactoside ^2^
7.15	350	463.1493	300.0692	Quercetin 3-*O*-glucoside ^2^
7.80	350	607.2236	300.1047	Quercetin-7-*O*-[3-hydroxy-3-methglutaroyl]hex
7.95	350	609.2332	301.1128	Quercetin-3-*O*-rutinoside (Rutin) ^2^

^1^ Experimental data. ^2^ Identified using corresponding authentic standards. Abbreviations: meth—methyl, hex—hexoside.

**Table 3 ijms-25-07572-t003:** The content of phenolic acids and flavanols (mg/100 mL)a identified in mint/nettle fresh infusion and kombucha brews during fermentation (average ± standard deviation; n = 3).

		Phenolic Acids																		Flavanols			
		Gallic Acid	Caffeic Acid	Protocatechuic Acid Hexoside	2,3-Dihydrobenzoic Acid	Digalloylglucose	Caffeoylquinic Acid	Sinapic Acid Pentoside	Digallic acid	3,5-Digalloylquinic Acid	p-Coumaroylquinic Acid	Digalloylglucose	Caffeoylquinic Acid	Caffeoylquinic Acid	Caffeoyl hexoside	p-Coumaroyl tartaric Acid	Salvianolic Acid B	Rosmarinic Acid	Sum of Phenolic Acids	(+)-Gallocatechin	(+)-Epigallocatechin	(+)-Gallocatechin 3-O-gallate	Sum of Flavanols
Fresh infusion		0.07 ± 0.00 ^ęf^	0.26 ± 0.01 ^n^	0.18 ± 0.01 ^i^	2.31 ± 0.18 ^ł^	0.13 ± 0.02 ^d^	3.22 ± 0.25 ^o^	0.18 ± 0.01 ^ę^	0.75 ± 0.02 ^a^	0.07 ± 0.01 ^ć^	0.59 ± 0.02 ^l^	0.07 ± 0.01 ^e^	5.18 ± 0.11 ^k^	1.62 ± 0.21 ^s^	0.89 ± 0.02 ^n^	4.51 ± 0.09 ^n^	0.72 ± 0.01 ^k^	1.43 ± 0.09 ^ń^	22.18 ± 0.55 ^ś^	0.33 ± 0.01 ^n^	0.65 ± 0.02 ^e^	0.75 ± 0.01 ^s^	1.72 ± 0.01 ^p^
K1	Day 0	0.07 ± 0.00 ^f^	0.14 ± 0.01 ^o^	0.22 ± 0.00 ^f^	2.12 ± 0.15 ^n^	Nd	4.42 ± 0.29 ^j^	Nd	Nd	Nd	0.65 ± 0.02 ^k^	Nd	5.36 ± 0.02 ^j^	2.44 ± 0.21 ^o^	2.66 ± 0.11 ^c^	4.44 ± 0.15 ^ń^	0.37 ± 0.01 ^ó^	0.99 ± 0.09 ^r^	23.88 ± 0.41 ^ó^	0.26 ± 0.01 ^ń^	2.33 ± 0.03 ^c^	0.29 ± 0.01 ^t^	2.87 ± 0.03 ^j^
	Day 2	0.09 ± 0.00 ^eę^	0.66 ± 0.02 ^d^	0.25 ± 0.01 ^ę^	2.55 ± 0.15 ^k^	Nd	5.78 ± 0.33 ^d^	Nd	Nd	Nd	1.04 ± 0.03 ^ę^	Nd	8.04 ± 0.09 ^c^	3.82 ± 0.19 ^i^	3.53 ± 0.15 ^b^	5.14 ± 0.21 ^j^	0.95 ± 0.02 ^h^	2.17 ± 0.08 ^f^	34.01 ± 0.57 ^g^	0.64 ± 0.02 ^j^	2.46 ± 0.02 ^b^	2.52 ± 0.01 ^d^	5.62 ± 0.03 ^b^
	Day 4	0.11 ± 0.01 ^ćd^	0.83 ± 0.02 ^b^	0.27 ± 0.02 ^e^	2.70 ± 0.19 ^i^	Nd	6.94 ± 0.41 ^ą^	Nd	Nd	Nd	1.33 ± 0.02 ^ą^	Nd	9.37 ± 0.12 ^ą^	4.12 ± 0.20 ^g^	3.99 ± 0.21 ^ą^	6.02 ± 0.32 ^f^	0.96 ± 0.02 ^h^	2.23 ± 0.10 ^ę^	38.87 ± 0.54 ^e^	0.78 ± 0.02 ^h^	2.87 ± 0.02 ^ą^	2.88 ± 0.02 ^ą^	6.53 ± 0.03 ^ą^
	Day 7	0.16 ± 0.01 ^c^	0.92 ± 0.01 ^a^	0.32 ± 0.02 ^d^	2.78 ± 0.21 ^h^	Nd	7.12 ± 0.25 ^a^	Nd	Nd	Nd	1.51 ± 0.02 ^a^	Nd	10.00 ± 0.51 ^a^	4.86 ± 0.22 ^e^	4.13 ± 0.24 ^a^	7.26 ± 0.33 ^ę^	0.99 ± 0.02 ^g^	2.31 ± 0.08 ^e^	42.36 ± 0.61 ^d^	0.89 ± 0.03 ^f^	3.21 ± 0.01 ^a^	3.01 ± 0.02 ^a^	7.11 ± 0.04 ^a^
	Day 11	0.09 ± 0.01 ^eę^	0.47 ± 0.01 ^ij^	0.19 ± 0.01 ^hi^	2.93 ± 0.18 ^g^	Nd	5.74 ± 0.36 ^e^	Nd	Nd	Nd	1.04 ± 0.03 ^ę^	Nd	7.94 ± 0.32 ^ć^	3.72 ± 0.30 ^j^	1.71 ± 0.12 ^ę^	4.94 ± 0.25 ^k^	1.00 ± 0.02 ^g^	2.08 ± 0.07 ^g^	31.84 ± 0.47 ^h^	1.30 ± 0.03 ^b^	0.57 ± 0.01 ^f^	2.43 ± 0.03 ^e^	4.29 ± 0.09 ^e^
	Day 14	0.10 ± 0.00 ^de^	0.50 ± 0.02 ^g^	0.17 ± 0.01 ^ijk^	3.52 ± 0.22 ^f^	Nd	5.77 ± 0.48 ^d^	Nd	Nd	Nd	1.08 ± 0.03 ^e^	Nd	7.87 ± 0.29 ^d^	3.83 ± 0.25 ^h^	1.82 ± 0.11 ^e^	1.78 ± 0.31 ^u^	1.05 ± 0.01 ^f^	2.19 ± 0.03 ^f^	29.67 ± 0.58 ^k^	1.46 ± 0.01 ^ą^	0.59 ± 0.01 ^ę^	2.74 ± 0.02 ^c^	4.79 ± 0.05 ^ć^
	Day 16	0.10 ± 0.01 ^de^	0.60 ± 0.03 ^ę^	0.18 ± 0.01 ^i^	4.11 ± 0.25 ^e^	Nd	6.37 ± 0.28 ^c^	Nd	Nd	Nd	1.20 ± 0.03 ^c^	Nd	9.14 ± 0.41 ^b^	4.16 ± 0.21 ^f^	2.06 ± 0.09 ^d^	5.67 ± 0.33 ^g^	1.20 ± 0.01 ^e^	2.48 ± 0.02 ^ć^	37.24 ± 0.60 ^ę^	1.79 ± 0.02 ^a^	0.74 ± 0.01 ^ć^	2.77 ± 0.01 ^b^	5.30 ± 0.08 ^c^
K2	Day 0	0.03 ± 0.00 ^h^	0.29 ± 0.02 ^m^	0.08 ± 0.00 ^ł^	1.76 ± 0.11 ^ń^	Nd	3.65 ± 0.21 ^n^	Nd	Nd	Nd	1.01 ± 0.01 ^f^	Nd	4.29 ± 0.28 ^n^	2.36 ± 0.11 ^ó^	1.48 ± 0.09 ^g^	2.57 ± 0.14 ^t^	1.25 ± 0.02 ^d^	2.38 ± 0.01 ^d^	21.14 ± 0.41 ^t^	1.04 ± 0.01 ^d^	0.27 ± 0.01 ^m^	1.25 ± 0.02 ^p^	2.56 ± 0.02 ^m^
	Day 2	0.04 ± 0.00 ^gh^	0.28 ± 0.01 ^m^	0.17 ± 0.01 ^ij^	1.48 ± 0.10 ^s^	Nd	4.13 ± 0.28 ^k^	Nd	Nd	Nd	1.09 ± 0.02 ^e^	Nd	4.69 ± 0.30 ^ł^	2.43 ± 0.14 ^o^	1.34 ± 0.11 ^j^	2.79 ± 0.10 ^p^	1.54 ± 0.02 ^ą^	2.63 ± 0.03 ^ą^	22.62 ± 0.33 ^r^	0.24 ± 0.01 ^o^	0.23 ± 0.02 ^ń^	1.19 ± 0.01 ^r^	1.65 ± 0.01 ^r^
	Day 4	0.04 ± 0.01 ^gh^	0.29 ± 0.02 ^m^	0.15 ± 0.01 ^k^	1.26 ± 0.09 ^ś^	Nd	4.02 ± 0.33 ^ł^	Nd	Nd	Nd	1.13 ± 0.02 ^d^	Nd	4.60 ± 0.27 ^m^	2.49 ± 0.10 ^ń^	1.40 ± 0.12 ^i^	2.75 ± 0.10 ^s^	1.45 ± 0.01 ^b^	2.59 ± 0.02 ^b^	22.17 ± 0.32 ^ś^	0.49 ± 0.01 ^l^	0.25 ± 0.01 ^n^	1.58 ± 0.01 ^ł^	2.32 ± 0.02 ^ń^
	Day 7	0.05 ± 0.01 ^gh^	0.36 ± 0.02 ^kl^	0.21 ± 0.02 ^fg^	1.76 ± 0.11 ^ń^	Nd	4.78 ± 0.21 ^h^	Nd	Nd	Nd	1.24 ± 0.01 ^b^	Nd	5.17 ± 0.22 ^k^	2.91 ± 0.12 ^ł^	1.48 ± 0.08 ^g^	3.21 ± 0.21 ^o^	1.78 ± 0.03 ^a^	3.07 ± 0.01 ^a^	26.01 ± 0.28 ^o^	0.35 ± 0.01 ^m^	0.27 ± 0.01 ^m^	1.57 ± 0.02 ^ł^	2.19 ± 0.02 ^oó^
	Day 11	0.03 ± 0.01 ^gh^	0.31 ± 0.01 ^ł^	0.09 ± 0.01 ^ł^	1.64 ± 0.10 ^ó^	Nd	3.99 ± 0.19 ^m^	Nd	Nd	Nd	1.09 ± 0.02 ^e^	Nd	4.61 ± 0.31 ^m^	2.52 ± 0.13 ^n^	1.55 ± 0.11 ^f^	2.77 ± 0.14 ^r^	1.35 ± 0.02 ^ć^	2.52 ± 0.03 ^c^	22.46 ± 0.45 ^s^	0.95 ± 0.02 ^ę^	0.29 ± 0.02 ^m^	1.36 ± 0.01 ^ó^	2.59 ± 0.03 ^ł^
	Day 14	0.04 ± 0.00 ^gh^	0.35 ± 0.01 ^l^	0.11 ± 0.00 ^l^	1.70 ± 0.12 ^o^	Nd	4.08 ± 0.32 ^l^	Nd	Nd	Nd	1.15 ± 0.01 ^ć^	Nd	4.82 ± 0.20 ^l^	2.57 ± 0.19 ^m^	1.56 ± 0.14 ^f^	2.86 ± 0.18 ^ó^	1.43 ± 0.01 ^c^	2.63 ± 0.01 ^ą^	23.29 ± 0.31 ^p^	0.85 ± 0.02 ^g^	0.29 ± 0.02 ^m^	1.42 ± 0.02 ^o^	2.55 ± 0.02 ^m^
	Day 16	0.05 ± 0.01 ^g^	0.11 ± 0.01 ^ó^	0.16 ± 0.01 ^jk^	1.53 ± 0.11 ^r^	Nd	3.59 ± 0.21 ^ń^	Nd	Nd	Nd	0.48 ± 0.01 ^ł^	Nd	3.65 ± 0.19 ^o^	1.72 ± 0.18 ^r^	0.63 ± 0.13 ^o^	2.73 ± 0.14 ^ś^	1.00 ± 0.02 ^g^	1.84 ± 0.01 ^j^	17.48 ± 0.29 ^w^	0.06 ± 0.00 ^p^	0.15 ± 0.01 ^o^	0.22 ± 0.01 ^u^	0.43 ± 0.01 ^ś^
K3	Day 0	0.12 ± 0.01 ^ć^	0.21 ± 0.02 ^ń^	0.41 ± 0.02 ^ć^	8.72 ± 0.31 ^a^	0.05 ± 0.01 ^f^	0.72 ± 0.01 ^t^	7.04 ± 0.25 ^e^	0.30 ± 0.01 ^d^	0.17 ± 0.01 ^b^	0.12 ± 0.01 ^ó^	0.52 ± 0.02 ^d^	0.04 ± 0.00 ^s^	3.84 ± 0.22 ^h^	1.43 ± 0.10 ^h^	7.52 ± 0.21 ^e^	0.51 ± 0.01 ^n^	Nd	31.71 ± 0.47 ^i^	0.34 ± 0.01 ^mn^	0.37 ± 0.01 ^l^	0.40 ± 0.01 ^ś^	1.11 ± 0.01 ^s^
	Day 2	0.20 ± 0.02 ^b^	0.37 ± 0.02 ^k^	0.78 ± 0.03 ^c^	5.71 ± 0.39 ^ą^	0.09 ± 0.01 ^ę^	1.12 ± 0.02 ^r^	11.87 ± 0.41 ^b^	0.47 ± 0.02 ^b^	0.31 ± 0.02 ^a^	0.17 ± 0.00 ^o^	1.01 ± 0.02 ^b^	0.05 ± 0.00 ^r^	6.59 ± 0.25 ^ć^	2.32 ± 0.17 ^ć^	12.70 ± 0.33 ^b^	0.79 ± 0.01 ^j^	1.78 ± 0.02 ^l^	46.32 ± 0.45 ^b^	0.54 ± 0.02 ^k^	0.52 ± 0.02 ^h^	2.17 ± 0.02 ^f^	3.23 ± 0.02 ^h^
	Day 4	0.23 ± 0.02 ^ą^	0.49 ± 0.02 ^gh^	0.80 ± 0.03 ^c^	4.74 ± 0.44 ^d^	0.14 ± 0.02 ^d^	1.18 ± 0.01 ^ó^	14.08 ± 0.35 ^ą^	0.50 ± 0.02 ^ą^	0.10 ± 0.01 ^c^	0.17 ± 0.01 ^o^	1.12 ± 0.03 ^ą^	0.08 ± 0.00 ^p^	7.82 ± 0.23 ^ą^	1.56 ± 0.12 ^f^	14.41 ± 0.21 ^a^	0.87 ± 0.02 ^i^	2.08 ± 0.01 ^g^	50.35 ± 0.32 ^ą^	0.71 ± 0.02 ^i^	0.58 ± 0.02 ^ęf^	2.34 ± 0.02 ^ę^	3.63 ± 0.02 ^ę^
	Day 7	0.21 ± 0.01 ^b^	0.63 ± 0.03 ^e^	1.17 ± 0.08 ^a^	4.99 ± 0.41 ^c^	0.14 ± 0.01 ^d^	1.14 ± 0.02 ^p^	14.30 ± 0.28 ^a^	0.47 ± 0.01 ^b^	0.25 ± 0.01 ^ą^	0.20 ± 0.01 ^ń^	1.29 ± 0.03 ^a^	0.10 ± 0.01 ^ó^	8.20 ± 0.33 ^a^	1.31 ± 0.15 ^k^	14.26 ± 0.24 ^ą^	0.66 ± 0.01 ^l^	1.94 ± 0.02 ^i^	51.25 ± 0.29 ^a^	1.15 ± 0.03 ^c^	0.73 ± 0.03 ^d^	2.56 ± 0.03 ^ć^	4.43 ± 0.02 ^d^
	Day 11	0.16 ± 0.01 ^c^	0.57 ± 0.03 ^f^	0.89 ± 0.04 ^b^	3.89 ± 0.32 ^ę^	0.12 ± 0.01 ^e^	0.76 ± 0.01 ^ś^	9.80 ± 0.11 ^d^	0.28 ± 0.01 ^e^	0.05 ± 0.00 ^d^	0.20 ± 0.01 ^ń^	0.84 ± 0.01 ^ć^	0.09 ± 0.01 ^óp^	5.52 ± 0.31 ^d^	0.84 ± 0.11 ^ń^	9.37 ± 0.36 ^d^	0.48 ± 0.02 ^ń^	1.22 ± 0.01 ^p^	35.05 ± 0.31 ^f^	0.85 ± 0.02 ^g^	0.44 ± 0.01 ^j^	1.76 ± 0.01 ^k^	3.04 ± 0.02 ^i^
	Day 14	0.20 ± 0.01 ^b^	0.77 ± 0.02 ^ć^	1.11 ± 0.09 ^ą^	5.45 ± 0.29 ^b^	0.14 ± 0.02 ^d^	0.84 ± 0.01 ^s^	11.50 ± 0.24 ^ć^	0.35 ± 0.01 ^c^	0.06 ± 0.00 ^d^	0.27 ± 0.00 ^m^	0.97 ± 0.01 ^c^	0.09 ± 0.01 ^óp^	6.83 ± 0.20 ^b^	0.83 ± 0.01 ^ń^	11.52 ± 0.35 ^c^	0.56 ± 0.02 ^m^	1.62 ± 0.01 ^m^	43.09 ± 0.37 ^c^	0.99 ± 0.01 ^e^	0.45 ± 0.02 ^ij^	2.07 ± 0.02 ^h^	3.51 ± 0.01 ^g^
	Day 16	0.21 ± 0.02 ^b^	0.89 ± 0.03 ^ą^	1.18 ± 0.09 ^a^	4.90 ± 0.35 ^ć^	0.16 ± 0.02 ^ć^	0.84 ± 0.02 ^s^	11.55 ± .030 ^c^	0.33 ± 0.02 ^ć^	0.06 ± 0.00 ^ć^	0.27 ± 0.01 ^m^	1.00 ± 0.01 ^b^	0.08 ± 0.01 ^p^	6.70 ± 0.34 ^c^	0.94 ± 0.05 ^m^	11.24 ± 0.10 ^ć^	0.56 ± 0.03 ^m^	1.55 ± 0.02 ^n^	42.45 ± 0.22 ^ć^	1.06 ± 0.01 ^ć^	0.54 ± 0.02 ^g^	1.94 ± 0.01 ^j^	3.55 ± 0.02 ^f^
K4	Day 0	0.07 ± 0.00 ^ęf^	0.15 ± 0.01 ^o^	0.20 ± 0.01 ^gh^	Nd	0.25 ± 0.01 ^ą^	4.52 ± 0.25 ^i^	Nd	Nd	Nd	0.22 ± 0.01 ^n^	Nd	4.15 ± 0.09 ^ń^	2.00 ± 0.15 ^p^	0.33 ± 0.02 ^s^	4.66 ± 0.21 ^ł^	0.42 ± 0.02 ^o^	0.96 ± 0.01 ^s^	17.91 ± 0.09 ^u^	0.10 ± 0.01 ^ó^	0.14 ± 0.01 ^o^	0.19 ± 0.01 ^w^	0.43 ± 0.02 ^ś^
	Day 2	0.10 ± 0.01 ^de^	0.46 ± 0.02 ^j^	0.24 ± 0.01 ^ę^	2.34 ± 0.12 ^l^	0.23 ± 0.01 ^b^	5.75 ± 0.21 ^e^	Nd	Nd	Nd	0.72 ± 0.02 ^j^	Nd	5.71 ± 0.10 ^i^	3.67 ± 0.14 ^l^	0.12 ± 0.01 ^ś^	4.81 ± 0.22 ^l^	0.66 ± 0.01 ^l^	1.40 ± 0.01 ^o^	26.20 ± 0.12 ^ń^	0.25 ± 0.01 ^ńo^	0.46 ± 0.01 ^i^	1.46 ± 0.02 ^n^	2.17 ± 0.01 ^ó^
	Day 4	0.07 ± 0.01 ^ęf^	0.48 ± 0.02 ^hi^	0.27 ± 0.02 ^e^	1.61 ± 0.12 ^p^	0.27 ± 0.02 ^a^	5.62 ± 0.33 ^ę^	Nd	Nd	Nd	0.77 ± 0.02 ^h^	Nd	6.11 ± 0.12 ^g^	3.70 ± 0.21 ^k^	1.19 ± 0.03 ^l^	5.47 ± 0.34 ^h^	1.07 ± 0.02 ^ę^	2.00 ± 0.02 ^h^	28.63 ± 0.08 ^ł^	0.25 ± 0.02 ^ńo^	0.36 ± 0.02 ^l^	2.11 ± 0.02 ^g^	2.72 ± 0.02 ^k^
	Day 7	0.28 ± 0.01 ^a^	0.81 ± 0.03 ^c^	0.32 ± 0.02 ^d^	2.94 ± 0.23 ^g^	0.23 ± 0.02 ^b^	6.47 ± 0.24 ^b^	Nd	Nd	Nd	0.78 ± 0.01 ^g^	Nd	6.61 ± 0.10 ^e^	4.36 ± 0.23 ^ę^	0.47 ± 0.02 ^ó^	5.41 ± 0.38 ^i^	0.78 ± 0.02 ^j^	1.70 ± 0.01 ^ł^	31.17 ± 0.21 ^j^	0.50 ± 0.01 ^l^	0.32 ± 0.01 ^ł^	1.66 ± 0.01 ^l^	2.47 ± 0.02 ^n^
	Day 11	0.21 ± 0.00 ^b^	0.63 ± 0.02 ^e^	0.27 ± 0.01 ^e^	2.29 ± 0.19 ^m^	0.20 ± 0.01 ^c^	5.47 ± 0.38 ^g^	Nd	Nd	Nd	0.71 ± 0.03 ^j^	Nd	6.15 ± 0.15 ^f^	3.67 ± 0.21 ^l^	0.39 ± 0.01 ^p^	4.57 ± 0.33 ^m^	0.60 ± 0.03 ^ł^	1.38 ± 0.02 ^ó^	26.53 ± 0.19 ^n^	0.43 ± 0.02 ^ł^	0.27 ± 0.02 ^m^	1.50 ± 0.02 ^m^	2.20 ± 0.01 ^o^
	Day 14	0.17 ± 0.01 ^c^	0.57 ± 0.02 ^f^	0.27 ± 0.02 ^e^	2.59 ± 0.14 ^j^	0.20 ± 0.01 ^c^	5.93 ± 0.14 ^ć^	Nd	Nd	Nd	0.73 ± 0.01 ^i^	Nd	5.75 ± 0.20 ^h^	3.81 ± 0.18 ^i^	0.34 ± 0.01 ^r^	4.81 ± 0.31 ^l^	0.66 ± 0.01 ^l^	1.42 ± 0.01 ^ń^	27.24 ± 0.20 ^m^	0.35 ± 0.02 ^m^	0.40 ± 0.03 ^k^	1.44 ± 0.01 ^ń^	2.20 ± 0.03 ^o^
	Day 16	0.08 ± 0.01 ^ęf^	0.48 ± 0.03 ^hi^	0.25 ± 0.02 ^ę^	2.54 ± 0.23 ^k^	0.26 ± 0.01 ^ą^	5.59 ± 0.25 ^f^	Nd	Nd	Nd	0.75 ± 0.02 h	Nd	6.49 ± 0.22 ^ę^	3.67 ± 0.24 ^l^	1.12 ± 0.02 ^ł^	5.15 ± 0.28 ^j^	0.95 ± 0.03 ^h^	1.82 ± 0.02 ^k^	29.13 ± 0.21 ^l^	0.34 ± 0.02 ^mn^	0.32 ± 0.01 ^ł^	2.03 ± 0.02 ^i^	2.69 ± 0.02^;^

Means of three independent analyses ± standard deviation; Values in the same columns followed by different letters are significantly different at *p* < 0.05 according to Duncan’s test. Abbreviations: K1—kombucha with 10% of sugar addition, fermented in 20 °C; K2—kombucha with 15% of sugar addition, fermented in 20 °C; K3—kombucha with 10% of sugar addition, fermented in 37 °C; K4—kombucha with 15% of sugar addition, fermented in 37 °C; Nd—not detected.

**Table 4 ijms-25-07572-t004:** The content of flavones, flavanones and flavonols (mg/100 mL) a identified mint/nettle fresh infusion and kombucha brews during fermentation (average ± standard deviation; n = 3).

	Flavones and Flavanones											Flavonols									
		Luteolin Diglucoside	Luteolin Diglucuronide	Apigenin 6,8-di-C-glucoside	Apigenin C-Hexoside-C-pentoside	Diosmin	Neoeriocitrin	Luteolin-7-O-rutinoside	Luteolin-7-O-glucuronide	Apigenin-7-O-rutinoside (Isorhoifolin)	Sum of Flavones and Flavanones	Dihydromyricetin	Quercetin-diglucoside	Isorhamnetin Dihexoside	Isorhamnetin Dihexoside	Quercetin-O-galloyl-glucoside	Quercetin 3-O-galactoside	Quercetin 3-O-glucoside	Quercetin-7-O-[3-hydroxy-3-methylglutaroyl]hexoside	Quercetin-3-O-rutinoside (Rutin)	Sum of Flavonols
Fresh infusion		Nd	0.43 ± 0.02 ^j^	29.18 ± 0.21 ^ń^	3.44 ± 0.12 ^t^	3.45 ± 0.15 ^s^	50.23 ± 1.48 ^t^	14.57 ± 0.58 ^w^	0.74 ± 0.02 ^m^	2.09 ± 0.05 ^ó^	104.13 ± 2.14 ^y^	0.75 ± 0.02 ^ł^	0.15 ± 0.01 ^ę^	0.04 ± 0.00 ^ę^	0.02 ± 0.00 ^ę^	0.02 ± 0.00 ^ć^	0.46 ± 0.01 ^ń^	0.35 ± 0.01 ^p^	1.87 ± 0.03 ^o^	0.26 ± 0.01 ^i^	3.90 ± 0.41 ^ś^
K1	Day 0	Nd	0.70 ± 0.03 ^b^	77.74 ± 0.45 ^c^	5.30 ± 0.14 ^ń^	3.98 ± 0.11 ^p^	66.77 ± 2.12 ^ł^	20.62 ± 1.01 ^m^	0.93 ± 0.03 ^k^	3.31 ± 0.04 ^ć^	179.35 ± 3.14 ^e^	0.92 ± 0.03 ^j^	Nd	Nd	Nd	Nd	0.94 ± 0.02 ^ę^	1.03 ± 0.03 ^j^	2.53 ± 0.09 ^i^	0.43 ± 0.02 ^d^	5.86 ± 1.23 ^ł^
	Day 2	Nd	0.67 ± 0.02 ^c^	41.16 ± 0.32 ^l^	9.13 ± 0.10 ^ę^	6.31 ± 0.21 ^e^	78.03 ± 3.15 ^e^	21.71 ± 0.98 ^h^	1.09 ± 0.02 ^g^	3.01 ± 0.05 ^h^	161.10 ± 3.33 ^h^	1.63 ± 0.03 ^c^	Nd	Nd	Nd	Nd	1.11 ± 0.10 ^ć^	1.36 ± 0.02 ^d^	2.24 ± 0.05 ^mn^	0.42 ± 0.03 ^de^	6.76 ± 0.99 ^g^
	Day 4	Nd	0.66 ± 0.02 ^c^	41.45 ± 0.33 ^j^	10.00 ± 0.09 ^d^	6.45 ± 0.18 ^d^	85.17 ± 3.33 ^b^	22.87 ± 1.12 ^c^	1.19 ± 0.03 ^e^	3.43 ± 0.01 ^c^	171.22 ± 2.85 ^ę^	1.78 ± 0.04 ^ą^	Nd	Nd	Nd	Nd	1.15 ± 0.09 ^c^	1.44 ± 0.02 ^c^	2.33 ± 0.10 ^l^	0.50 ± 0.02 ^b^	7.20 ± 1.05 ^e^
	Day 7	Nd	0.67 ± 0.03 ^c^	41.57 ± 0.45 ^i^	10.99 ± 0.11 ^b^	6.66 ± 0.25 ^ć^	99.12 ± 4.19 ^a^	24.15 ± 1.22 ^b^	1.35 ± 0.03 ^ć^	3.50 ± 0.02 ^ą^	188.01 ± 3.45 ^b^	1.99 ± 0.02 ^a^	Nd	Nd	Nd	Nd	1.23 ± 0.02 ^ą^	1.53 ± 0.01 ^b^	2.79 ± 0.06 ^f^	5.51 ± 0.12 ^a^	13.05 ± 1.26 ^ą^
	Day 11	Nd	0.66 ± 0.02 ^c^	41.24 ± 0.62 ^k^	8.53 ± 0.15 ^g^	6.05 ± 0.20 ^g^	76.72 ± 4.56 ^ę^	21.34 ± 1.28 ^i^	1.06 ± 0.02 ^h^	3.12 ± 0.04 ^ę^	158.71 ± 5.17 ^i^	1.59 ± 0.03 ^ć^	Nd	Nd	Nd	Nd	1.04 ± 0.08 ^d^	1.34 ± 0.03 ^e^	2.68 ± 0.07 ^g^	0.46 ± 0.02 ^ć^	7.10 ± 2.15 ^ę^
	Day 14	Nd	0.66 ± 0.03 ^c^	43.23 ± 0.21 ^h^	9.67 ± 0.21 ^e^	6.29 ± 0.17 ^e^	80.05 ± 3.51 ^c^	22.52 ± 1.32 ^d^	1.13 ± 0.02 ^f^	3.32 ± 0.03 ^ć^	166.86 ± 4.04 ^g^	1.66 ± 0.05 ^b^	Nd	Nd	Nd	Nd	1.12 ± 0.02 ^ć^	1.40 ± 0.02 ^ć^	2.83 ± 0.05 ^ę^	0.48 ± 0.03 ^c^	7.48 ± 1.48 ^d^
	Day 16	Nd	0.76 ± 0.03 ^ą^	46.71 ± 0.33 ^g^	10.10 ± 0.32 ^ć^	6.85 ± 0.30 ^c^	89.02 ± 3.29 ^ą^	25.32 ± 1.02 ^ą^	1.31 ± 0.01 ^d^	3.62 ± 0.02 ^a^	183.69 ± 3.64 ^c^	1.78 ± 0.03 ^ą^	Nd	Nd	Nd	Nd	1.28 ± 0.01 ^a^	1.54 ± 0.02 ^b^	3.21 ± 0.09 ^b^	0.53 ± 0.03 ^ą^	8.34 ± 2.07 ^c^
K2	Day 0	Nd	0.62 ± 0.02 ^d^	10.28 ± 0.21 ^y^	4.74 ± 0.21 ^p^	5.44 ± 0.24 ^m^	66.30 ± 3.27 ^m^	20.74 ± 1.58 ^l^	1.31 ± 0.02 ^d^	2.92 ± 0.03 ^k^	112.34 ± 2.58 ^t^	0.95 ± 0.05 ^i^	Nd	Nd	Nd	Nd	0.68 ± 0.01 ^k^	0.66 ± 0.02 ^ó^	2.87 ± 0.09 ^e^	0.39 ± 0.04 ^ę^	5.56 ± 0.98 ^ó^
	Day 2	Nd	0.71 ± 0.03 ^b^	13.51 ± 0.18 ^ś^	3.13 ± 0.25 ^u^	5.86 ± 0.20 ^h^	68.79 ± 2.98 ^k^	21.85 ± 1.06 ^g^	1.52 ± 0.03 ^ą^	2.95 ± 0.02 ^j^	118.32 ± 3.22 ^r^	0.93 ± 0.03 ^ij^	Nd	Nd	Nd	Nd	0.61 ± 0.02 ^n^	0.74 ± 0.01 ^n^	3.06 ± 0.11 ^ć^	0.36 ± 0.02 ^f^	5.70 ± 2.15 ^m^
	Day 4	Nd	0.70 ± 0.03 ^b^	12.45 ± 0.15 ^t^	4.19 ± 0.17 ^ś^	5.83 ± 0.30 ^i^	70.19 ± 3.64 ^i^	22.11 ± 1.44 ^ę^	1.45 ± 0.02 ^b^	2.99 ± 0.02 ^i^	119.90 ± 4.12 ^o^	0.95 ± 0.01 ^i^	Nd	Nd	Nd	Nd	0.65 ± 0.02 ^lł^	0.71 ± 0.01 ^ń^	2.95 ± 0.13 ^d^	0.37 ± 0.03 ^f^	5.63 ± 1.33 ^ń^
	Day 7	Nd	0.81 ± 0.04 ^a^	15.02 ± 0.21 ^s^	4.28 ± 0.15 ^s^	7.00 ± 0.29 ^b^	79.57 ± 3.30 ^ć^	25.65 ± 1.95 ^a^	1.90 ± 0.04 ^a^	3.45 ± 0.01 ^b^	137.68 ± 2.87 ^l^	0.99 ± 0.02 ^h^	Nd	Nd	Nd	Nd	0.75 ± 0.03 ^j^	0.84 ± 0.03 ^ł^	3.52 ± 0.02 ^ą^	0.44 ± 0.02 ^d^	6.53 ± 2.09 ^h^
	Day 11	Nd	0.62 ± 0.02 ^d^	11.68 ± 0.10 ^w^	4.67 ± 0.26 ^r^	5.72 ± 0.35 ^k^	69.83 ± 2.88 ^j^	21.90 ± 1.25 ^f^	1.35 ± 0.03 ^ć^	3.05 ± 0.03 ^g^	118.82 ± 3.85 ^p^	0.90 ± 0.03 ^k^	Nd	Nd	Nd	Nd	0.64 ± 0.03 ^ł^	0.69 ± 0.02 ^o^	2.96 ± 0.09 ^d^	0.37 ± 0.02 ^f^	5.56 ± 1.28 ^ó^
	Day 14	Nd	0.70 ± 0.03 ^b^	12.36 ± 0.11 ^u^	4.89 ± 0.31 ^o^	6.14 ± 0.27 ^f^	71.90 ± 2.12 ^g^	22.64 ± 0.99 ^ć^	1.51 ± 0.02 ^ą^	3.18 ± 0.01 ^e^	123.31 ± 3.17 ^m^	1.00 ± 0.01 ^h^	Nd	Nd	Nd	Nd	0.69 ± 0.01 ^k^	0.79 ± 0.01 ^m^	3.14 ± 0.05 ^c^	0.42 ± 0.03 ^de^	6.04 ± 3.05 ^k^
	Day 16	Nd	0.70 ± 0.04 ^b^	40.97 ± 0.32 ł	1.45 ± 0.17 ^w^	2.62 ± 0.24 ^ś^	63.31 ± 3.87 ^ń^	20.69 ± 1.34 ^ł^	1.39 ± 0.05 ^c^	3.09 ± 0.02 ^f^	134.21 ± 2.69 ^ł^	0.80 ± 0.02 ^l^	Nd	Nd	Nd	Nd	Nd	0.06 ± 0.04 ^r^	2.96 ± 0.14 ^d^	0.30 ± 0.01 ^h^	4.12 ± 1.47 ^r^
K3	Day 0	0.03 ± 0.00 ^bc^	0.35 ± 0.02 ^l^	48.90 ± 0.21 ^f^	4.81 ± 0.32 ^ó^	3.64 ± 0.19 ^r^	38.21 ± 1.99 ^w^	10.59 ± 1.54 ^y^	0.48 ± 0.04 ^ń^	1.69 ± 0.01 ^p^	108.70 ± 1.99 ^w^	Nd	0.85 ± 0.02 ^e^	0.26 ± 0.02 ^e^	0.09 ± 0.01 ^ć^	0.03 ± 0.00 ^c^	0.52 ± 0.02 ^n^	0.77 ± 0.03 ^m^	1.34 ± 0.09 ^p^	0.22 ± 0.01 ^k^	4.07 ± 1.09 ^s^
	Day 2	0.06 ± 0.00 ^aą^	0.54 ± 0.02 ^ę^	76.84 ± 0.45 ^ć^	6.91 ± 0.44 ^j^	5.74 ± 0.20 ^j^	59.92 ± 2.94 ^o^	16.65 ± 1.33 ^s^	0.71 ± 0.02 ^n^	2.66 ± 0.02 ^l^	170.03 ± 2.54 ^f^	Nd	1.37 ± 0.03 ^ć^	0.44 ± 0.02 ^d^	0.13 ± 0.01 ^c^	0.06 ± 0.01 ^aą^	0.77 ± 0.02 ^i^	1.04 ± 0.02 ^i^	2.07 ± 0.08 ^ń^	0.28 ± 0.02 ^h^	6.15 ± 2.14 ^j^
	Day 4	0.04 ± 0.01 ^b^	0.64 ± 0.03 ^ć^	87.87 ± 0.54 ^ą^	11.01 ± 0.29 ^ą^	7.03 ± 0.25 ^ą^	76.00 ± 3.38 ^f^	21.13 ± 1.62 ^j^	1.06 ± 0.03 ^h^	3.27 ± 0.02 ^d^	208.05 ± 3.69 ^ą^	Nd	1.65 ± 0.03 ^ą^	0.71 ± 0.03 ^ą^	0.19 ± 0.02 ^ą^	0.05 ± 0.00 ^aąb^	1.20 ± 0.01 ^b^	1.66 ± 0.02 ^ą^	2.66 ± 0.06 ^h^	0.43 ± 0.01 ^d^	8.55 ± 3.08 ^b^
	Day 7	0.06 ± 0.00 ^a^	0.67 ± 0.02 ^c^	97.33 ± 0.66 ^a^	12.61 ± 0.14 ^a^	7.40 ± 0.35 ^a^	79.29 ± 2.65 ^d^	22.17 ± 1.48 ^e^	0.34 ± 0.01 ^o^	0.62 ± 0.03 ^r^	220.50 ± 2.75 ^a^	Nd	1.84 ± 0.02 ^a^	0.78 ± 0.01 ^a^	0.24 ± 0.02 ^a^	0.06 ± 0.01 ^a^	1.24 ± 0.02 ^ą^	1.86 ± 0.03 ^a^	9.27 ± 0.17 ^a^	0.40 ± 0.03 ^eę^	15.70 ± 2.58 ^a^
	Day 11	0.03 ± 0.00 ^bc^	0.42 ± 0.01 ^k^	62.73 ± 0.41 ^e^	7.60 ± 0.27 ^h^	4.68 ± 0.31 ^ń^	54.46 ± 3.75 ^ś^	14.76 ± 1.09 ^t^	0.12 ± 0.01 ^p^	2.35 ± 0.04 ^ń^	147.14 ± 2.89 ^j^	0.07 ± 0.01 ^m^	1.31 ± 0.01 ^d^	0.55 ± 0.02 ^ć^	0.04 ± 0.00 ^e^	0.05 ± 0.00 ^b^	0.90 ± 0.03 ^f^	1.20 ± 0.01 ^g^	1.77 ± 0.19 ^ó^	0.32 ± 0.02 ^g^	6.20 ± 3.09 ^i^
	Day 14	0.03 ± 0.00 ^c^	0.51 ± 0.03 ^fg^	76.37 ± 0.55 ^d^	8.88 ± 0.33 ^f^	6.16 ± 0.29 ^ę^	67.09 ± 3.58 ^l^	18.97 ± 1.21 ^n^	0.18 ± 0.01 ^ó^	2.96 ± 0.01 ^j^	181.15 ± 3.64 ^d^	0.04 ± 0.00 ^n^	1.60 ± 0.02 ^b^	0.64 ± 0.03 ^c^	0.15 ± 0.01 ^b^	0.05 ± 0.01 ^aąb^	1.11 ± 0.02 ^ć^	1.35 ± 0.04 ^d^	2.40 ± 0.13 ^j^	0.42 ± 0.01 ^de^	7.76 ± 2.68 ^ć^
	Day 16	0.05 ± 0.00 ^ą^	0.48 ± 0.02 ^h^	78.75 ± 0.28 ^b^	10.49 ± 0.35 ^c^	6.05 ± 0.30 ^g^	66.09 ± 4.01 ^n^	18.33 ± 1.55 ^ń^	Nd	2.91 ± 0.02 ^k^	183.16 ± 4.08 ^ć^	0.06 ± 0.01 ^mn^	1.55 ± 0.03 ^c^	0.65 ± 0.01 ^b^	0.08 ± 0.01 ^d^	0.05 ± 0.00 ^ąb^	1.12 ± 0.01 ^ć^	1.29 ± 0.02 ^f^	2.31 ± 0.11 ^ł^	0.36 ± 0.02 ^f^	7.47 ± 3.47 ^d^
K4	Day 0	Nd	0.49 ± 0.02 ^gh^	53.93 ± 0.41 ^ę^	0.44 ± 0.18 ^y^	1.05 ± 0.01 ^t^	46.37 ± 3.99 ^u^	14.71 ± 1.30 ^u^	0.75 ± 0.03 ^ł^	2.28 ± 0.01 ^o^	120.04 ± 4.61 ^ń^	0.94 ± 0.01 ^ij^	Nd	Nd	Nd	Nd	Nd	Nd	2.06 ± 0.02	0.24 ± 0.01 ^j^	3.24 ± 2.19 ^t^
	Day 2	Nd	0.46 ± 0.03 ^i^	23.66 ± 0.21 ^r^	5.70 ± 0.24 ^m^	4.44 ± 0.19 ^o^	56.78 ± 2.77 ^s^	16.58 ± 1.47 ^ś^	0.87 ± 0.02 ^l^	2.39 ± 0.02 ^n^	110.87 ± 3.99 ^u^	1.26 ± 0.02 ^g^	Nd	Nd	Nd	Nd	0.73 ± 0.02 ^j^	0.89 ± 0.04 ^l^	2.23 ± 0.03 ^n^	0.34 ± 0.02 ^g^	5.44 ± 1.29 ^p^
	Day 4	Nd	0.57 ± 0.03 ^e^	29.10 ± 0.10 ^o^	6.59 ± 0.25 ^k^	5.64 ± 0.23 ^l^	59.66 ± 2.59 ^ó^	17.62 ± 1.28 ^o^	1.17 ± 0.02 ^ę^	2.45 ± 0.03 ^ł^	122.79 ± 3.71 ^n^	1.44 ± 0.01 ^e^	Nd	Nd	Nd	Nd	0.86 ± 0.02 ^g^	1.01 ± 0.03 ^k^	2.38 ± 0.09 ^k^	0.30 ± 0.02 ^h^	5.98 ± 2.66 ^l^
	Day 7	Nd	0.62 ± 0.02 ^ćd^	32.90 ± 0.41 ^m^	7.15 ± 0.33 ^i^	5.51 ± 0.20 ^ł^	70.51 ± 3.61 ^h^	20.81 ± 1.41 ^k^	1.36 ± 0.01 ^ć^	3.43 ± 0.01 ^c^	142.31 ± 2.15 ^k^	1.57 ± 0.02 ^d^	Nd	Nd	Nd	Nd	1.02 ± 0.01 ^e^	1.31 ± 0.02 ^ę^	2.65 ± 0.05 ^h^	0.37 ± 0.03	6.92 ± 3.47 ^f^
	Day 11	Nd	0.47 ± 0.03 ^i^	26.81 ± 0.33 ^ó^	5.77 ± 0.19 ^ł^	4.42 ± 0.19 ^ó^	59.10 ± 3.12 ^p^	17.21 ± 1.39 ^p^	0.93 ± 0.03 ^k^	2.40 ± 0.03 ^n^	117.10 ± 3.65 ^ś^	1.31 ± 0.02 ^ę^	Nd	Nd	Nd	Nd	0.82 ± 0.03 ^h^	1.01 ± 0.04 ^k^	2.26 ± 0.03 ^m^	0.26 ± 0.01 ^i^	5.65 ± 2.85 ^n^
	Day 14	Nd	0.46 ± 0.01 ^i^	26.65 ± 0.21 ^p^	6.04 ± 0.14 ^l^	4.40 ± 0.36 ^ó^	59.65 ± 2.99 ^ó^	17.29 ± 1.30 ^ó^	0.98 ± 0.02 ^j^	2.43 ± 0.02 ^m^	117.90 ± 4.23 ^s^	1.31 ± 0.03 ^ę^	Nd	Nd	Nd	Nd	0.66 ± 0.02 ^l^	1.09 ± 0.01 ^h^	2.33 ± 0.14 ^lł^	0.32 ± 0.02 ^g^	5.71 ± 1.39 ^m^
	Day 16	Nd	0.51 ± 0.02 ^f^	29.37 ± 0.17 ^n^	5.67 ± 0.31 ^n^	5.35 ± 0.28 ^n^	58.10 ± 3.14 ^r^	17.14 ± 1.08 ^r^	1.00 ± 0.03 ^i^	2.43 ± 0.02 ^łm^	119.56 ± 2.58 ^ó^	1.29 ± 0.01 ^f^	Nd	Nd	Nd	Nd	0.80 ± 0.01 ^h^	1.02 ± 0.02 ^jk^	2.25 ± 0.09 ^m^	0.25 ± 0.03 ^ij^	5.61 ± 2.52 ^o^

Means of three independent analyses ± standard deviation; Values in the same columns followed by different letters are significantly different at *p* < 0.05 according to Duncan’s test. Abbreviations: K1—kombucha with 10% of sugar addition, fermented in 20 °C; K2—kombucha with 15% of sugar addition, fermented in 20 °C; K3—kombucha with 10% of sugar addition, fermented in 37 °C; K4—kombucha with 15% of sugar addition, fermented in 37 °C; Nd—not detected.

**Table 5 ijms-25-07572-t005:** Antioxidant capacity (ABTS, FRAP, DPPH), anti-inflammatory (COX1, COX2), antidiabetic (anti-α-amylase, and-α-glucosidase), and anti-aging (acetylcholinesterase, butyrylcholinesterase) activities of mint/nettle infusion and kombucha brews during fermentation (average ± standard deviation; n = 3).

	Days	ABTS	FRAP	DPPH	α-Glucosidase	α-Amylase	Acetylcholinesterase	Butyrylcholinesterase	COX 1	COX 2
		µM Tx/100 mL			[% of Inhibition]					
Fresh infusion		78.44 ± 1.21 ^r^	61.76 ± 1.80 ^ś^	45.88 ± 1.12 ^u^	64.43 ± 2.51 ^e^	77.15 ± 4.63 ^ć^	59.03 ± 3.15 ^d^	40.42 ± 1.88 ^d^	33.87 ± 2.87 ^d^	26.69 ± 3.49 ^ć^
K1	Day 0	104.71 ± 2.55 ^i^	101.50 ± 2.24 ^ł^	75.49 ± 2.04 ^k^	65.20 ± 4.00 ^d^	79.17 ± 4.26 ^b^	59.56 ± 2.34 ^ć^	42.29 ± 3.12 ^ć^	27.53 ± 1.09 ^bc^	27.53 ± 1.09 ^bc^
	Day 2	106.44 ± 1.00 ^h^	106.76 ± 3.30 ^k^	77.06 ± 2.05 ^j^						
	Day 4	107.42 ± 0.83 ^g^	110.08 ± 6.26 ^i^	78.4 ± 1.88 ^h^						
	Day 7	83.32 ± 1.34 ^p^	77.38 ± 7.78 ^o^	57.95 ± 1.44 ^ń^	65.80 ± 3.52 ^c^	79.22 ± 4.12 ^b^	59.56 ± 1.11 ^ć^	42.30 ± 2.11 ^ć^	27.53 ± 1.57 ^bc^	27.53 ± 1.57 ^bc^
	Day 11	95.34 ± 1.32 ^n^	98.38 ± 3.86 ^m^	65.42 ± 3.73 ^n^						
	Day 14	71.49 ± 0.86 ^u^	71.14 ± 1.62 ^r^	47.37 ± 1.17 ^t^						
	Day 16	90.15 ± 1.51 ^ń^	93.78 ± 4.69 ^n^	57.2 ± 1.83 ^o^	70.21 ± 5.28 ^a^	83.25 ± 4.04 ^a^	61.15 ± 1.65 ^c^	58.48 ± 4.07 ^ą^	27.65 ± 0.70 ^bc^	27.65 ± 0.70 ^bc^
K2	Day 0	123.37 ± 1.19 ^a^	166.98 ± 5.76 ^ą^	95.50 ± 2.67 ^b^	53.91 ± 4.51 ^h^	71.14 ± 4.63 ^f^	59.56 ± 1.33 ^ć^	42.29 ± 3.21 ^ć^	27.53 ± 1.15 ^bc^	27.53 ± 1.15 ^bc^
	Day 2	122.93 ± 1.08 ^ą^	165.81 ± 2.08 ^b^	104.98 ± 2.86 ^ą^						
	Day 4	121.7 ± 1.01 ^b^	162.18 ± 2.06 ^c^	106.71 ± 2.66 ^a^						
	Day 7	119.09 ± 0.86 ^c^	169.96 ± 2.79 ^a^	95.19 ± 3.34 ^c^	55.45 ± 35.95 ^f^	72.19 ± 4.29 ^e^	59.57 ± 2.16 ^ć^	42.29 ± 3.19 ^ć^	27.49 ± 3.00 ^c^	27.49 ± 3.00 ^c^
	Day 11	117.96 ± 0.82 ^ć^	158.59 ± 7.36 ^e^	91.26 ± 1.34 ^ę^						
	Day 14	116.35 ± 1.06 ^e^	159.98 ± 3.41 ^ć^	94.13 ± 3.45 ^d^						
	Day 16	115.85 ± 2.02 ^ę^	159.47 ± 1.15 ^d^	92.12 ± 3.33 ^e^	65.98 ± 5.26 ^b^	77.49 ± 3.88 ^c^	63.58 ± 0.71 ^b^	55.62 ± 4.66 ^b^	27.89 ± 2.78 ^ą^	27.89 ± 2.78 ^ą^
K3	Day 0	87.28 ± 1.10 ^ó^	71.82 ± 1.60 ^p^	67.23 ± 2.22 ^m^	65.20 ± 2.98 ^d^	79.17 ± 3.67 ^b^	59.56 ± 1.45 ^ć^	42.29 ± 3.01 ^ć^	27.53 ± 1.76 ^bc^	27.53 ± 1.76 ^bc^
	Day 2	71.65 ± 3.63 ^u^	60.51 ± 0.88 ^t^	53.47 ± 2.88 ^ó^						
	Day 4	74.7 ± 1.56 ^ś^	55.13 ± 1.18 ^w^	51.5 ± 1.59 ^r^						
	Day 7	70.01 ± 1.43 ^w^	67.51 ± 1.97 ^s^	48.43 ± 2.39 ^ś^	65.52 ± 2.45 ^ć^	79.30 ± 5.45 ^b^	59.56 ± 1.04 ^ć^	42.28 ± 3.01 ^ć^	27.50 ± 1.55 ^c^	27.50 ± 1.55 ^c^
	Day 11	89.08 ± 1.42 ^o^	59.32 ± 1.45 ^u^	53.55 ± 2.41 ^ó^						
	Day 14	73.59 ± 2.89 ^t^	77.01 ± 1.96 ^ó^	51.11 ± 1.12 ^s^						
	Day 16	75.99 ± 1.57 ^s^	77.77 ± 1.24 ^ń^	51.89 ± 2.11 ^p^	68.12 ± 4.07 ^ą^	82.45 ± 5.42 ^ą^	65.01 ± 4.60 ^a^	68.57 ± 1.63 ^a^	29.15 ± 1.02 ^a^	29.15 ± 1.02 ^a^
K4	Day 0	116.69 ± 2.00 ^d^	104.39 ± 2.56 ^l^	94.44 ± 2.29 ^ć^	53.91 ± 2.21 ^h^	71.14 ± 4.17 ^f^	59.56 ± 2.01 ^ć^	42.29 ± 3.01 ^ć^	27.53 ± 1.44 ^bc^	27.53 ± 1.44 ^bc^
	Day 2	115.38 ± 3.95 ^f^	108.07 ± 2.45 ^j^	84.57 ± 1.56 ^f^						
	Day 4	101.09 ± 1.95 ^l^	104.44 ± 2.57 ^l^	75.45 ± 2.14 ^k^						
	Day 7	98.74 ± 5.52 ^m^	115.95 ± 1.31 ^g^	72.81 ± 2.84 ^l^	54.97 ± 1.49 ^g^	71.58 ± 3.63 ^ę^	59.55 ± 1.21 ^ć^	42.30 ± 1.59 ^ć^	27.49 ± 2.16 ^c^	27.49 ± 2.16 ^c^
	Day 11	101.38 ± 3.89 ^k^	114.25 ± 2.49 ^h^	72.46 ± 2.01 ^ł^						
	Day 14	101.98 ± 3.26 ^j^	122.48 ± 2.82 ^f^	78.67 ± 3.13 ^g^						
	Day 16	100.78 ± 2.89 ^ł^	126.67 ± 3.21 ^ę^	77.53 ± 3.48 ^i^	58.12 ± 2.32 ^ę^	74.78 ± 4.01 ^d^	64.15 ± 2.39 ^ą^	52.12 ± 2.43 ^c^	27.70 ± 1.93 ^b^	27.70 ± 1.93 ^b^

Means of three independent analyses ± standard deviation; Values in the same columns followed by different letters are significantly different at *p* < 0.05 according to Duncan’s test; Abbreviations: ABTS—2,2′-Azino-bis(3-ethylbenzothiazoline-6-sulfonic acid); DPPH—2,2-Diphenyl-1-(2,4,6-trinitrophenyl)hydrazyl; FRAP—ferric reducing antioxidant power assay; K1—kombucha with 10% of sugar addition, fermented in 20 °C; K2—kombucha with 15% of sugar addition, fermented in 20 °C; K3—kombucha with 10% of sugar addition, fermented in 37 °C; K4—kombucha with 15% of sugar addition, fermented in 37 °C.

## Data Availability

Data will be made available on request.

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
