# Peer review of "Innovative Analogs of Unpasteurized Kombucha Beverages: Comparative Analysis of Mint/Nettle Kombuchas, Considering Their Health-Promoting Effect, Polyphenolic Compounds and Chemical Composition"

_ijms, 2024, doi:10.3390/ijms25147572_

Round 1

Reviewer 1 Report

Comments and Suggestions for Authors

The manuscript provides clear and relevant information on the influence of fermentation parameters on the quality of kombucha drinks made from mint/nettle infusions. The text is well-structured, making it easy to follow the progression of ideas. The references cited are predominantly recent publications within the last 5 years, contributing to the relevance of the study. There are no indications of excessive self-citations. The scientific soundness of the manuscript appears to be robust, with an appropriate experimental design to test the hypothesis. The details provided in the methods section suggest that the results should be reproducible. Figures, tables, and data present are relevant to the study and are generally easy to interpret. The data is appropriately analyzed, and the presentation is consistent throughout the manuscript. The conclusions drawn in the manuscript align with the presented evidence and arguments. The discussion on the bioactive qualities of kombucha beverages and their correlation with organic acids is well-supported by the results obtained from chemometric investigations. The need for further research on raw materials, fermentation conditions, and ingredient interactions is appropriately emphasized based on the study's findings.

1.      I recommend that the authors incorporate information in the introduction section regarding the biologically active components found in mint and nettle. This addition would provide valuable context and background information for the readers, enhancing the overall understanding of the study.

2.      In the materials and methods section, it is imperative to provide details about the starter culture used and the specific growth medium in which it was cultivated.

3.      In the Results and Discussion section, it is important to note that Figures 1, 2 and 3 are currently missing standard deviation values, which should be included for completeness and accuracy.

In conclusion, the manuscript demonstrates a thorough investigation into the impact of fermentation parameters on kombucha quality using mint/nettle infusions. The study's structure, scientific rigor, and conclusions reflect a well-conducted research effort. Some minor improvements in data could further enhance the overall quality of the manuscript.

Author Response

Dear Reviewer, thank you for your valuable comments. Below are the responses and a list of changes made:

  1. I recommend that the authors incorporate information in the introduction section regarding the biologically active components found in mint and nettle. This addition would provide valuable context and background information for the readers, enhancing the overall understanding of the study.

Reply: Information has been added

  1. In the materials and methods section, it is imperative to provide details about the starter culture used and the specific growth medium in which it was cultivated.

Reply: Information has been added in the materials and methods section

  1. In the Results and Discussion section, it is important to note that Figures 1, 2 and 3 are currently missing standard deviation values, which should be included for completeness and accuracy.

Reply: Corrected. Due to the low readability of the figure after adding standard deviation values, it was decided to convert the figures into tables. Additionally, homogeneous groups were added to the results

Reviewer 2 Report

Comments and Suggestions for Authors

The present manuscript is well-structured and organized.  Presents suggestive tables and figures and suggestive references.

The aspect that can be improved for the present manuscript is the justification in the introduction of the choice of the two species:  Mentha piperita and Urtica dioica

Author Response

Reply: Dear Reviewer, thank you for your valuable comments. The following changes were made to the manuscript:

The information about vegetal material and the information about it’s bio-health properties and bioactive compounds have been added to manuscript.

Reviewer 3 Report

Comments and Suggestions for Authors

The present reseach investigates the antioxidant activity and other health benefits of several preparations of kombucha tested along the fermation time, the added sugar and the temperature of the reaction.

The Abstract is concise and resumes the work done.

The introduction presents a correct bibliographic review and details the problem in an argued form.

The results are well presented but also discussed with appropriated litterature. Only figures should be completed with the error bars and also the legends should be completed.

The discussion is really interesting.

The material and methods are clear, but should be completed in the identification of compounds by UPLC-PDA-Q/Tof-MS. Also is needed more details on the initial vegetal material used and the first brew used to lunch the process.

THe conclusions are online to the results obtained.

One reference should be changer and be remplaced by others.

Other minor comments in the document attached

Author Response

Reply: Dear Reviewer, thank you for your valuable comments. The following changes were made to the manuscript:

  1. Figures: Due to the low readability of the figure after adding standard deviation values, it was decided to convert the figures into tables. Additionally, homogeneous groups were added to the results
  2. Details of UPLC-PDA-Q/Tof-MS method have been added to ‘Material and Methods’ section. We also added information about starter culture and vegetal material.
  3. Reference has been changed
  4. Lines 189-191 – citation has been added
  5. Lines 409 and 411 – Camellia sinensis – has been changed to italic
  6. Figure 4 (PCA) – legend has been changed

Round 2

Reviewer 3 Report

Comments and Suggestions for Authors

The authors have now provided a corrected version of the document, were particular attention have been paid to the introduction, adding two supplementary paragraphs on the two main constituents of the Kombucha drinks, then figures have been changed to tables to better show the error and the statistical analysis and the other corrections have been done,

the only thing missing is in tables 1,3, 4 & 5 add in the legend what is K1 to K4 as it has been done in figure 1. Then it'll be ready for publication

Author Response

Dear Reviewer,

Thank you for your comments. Abbreviations have been added under the tables as suggested